# Uni-X: Mitigating Modality Conflict with a Two-End-Separated Architecture for Unified Multimodal Models

**Jitai Hao**[1,*]  **Hao Liu**[2,*]  **Xinyan Xiao**[2]  **Qiang Huang**[1,†]  **Jun Yu**[1,†]

[1]School of Intelligence Science and Engineering, Harbin Institute of Technology (Shenzhen)
[2]Baidu Inc.
`jitaihao@outlook.com`, `{huangqiang, yujun}@hit.edu.cn`
`{liuhao24, xiaoxinyan}@baidu.com`
[*]Equal contribution.    [†]Corresponding authors.

## Abstract

Unified Multimodal Models (UMMs) built on shared autoregressive (AR) transformers are attractive for their architectural simplicity. However, we identify a critical limitation: when trained on multimodal inputs, modality-shared transformers suffer from severe gradient conflicts between vision and text, particularly in shallow and deep layers. We trace this issue to the fundamentally different low-level statistical properties of images and text, while noting that conflicts diminish in middle layers where representations become more abstract and semantically aligned. To overcome this challenge, we propose Uni-X, a two-end-separated, middle-shared architecture. Uni-X dedicates its initial and final layers to modality-specific processing, while maintaining shared parameters in the middle layers for high-level semantic fusion. This X-shaped design not only eliminates gradient conflicts at both ends but also further alleviates residual conflicts in the shared layers. Extensive experiments validate the effectiveness of Uni-X. Under identical training conditions, Uni-X achieves superior training efficiency compared to strong baselines. When scaled to 3B parameters with larger training data, Uni-X matches or surpasses 7B AR-based UMMs, achieving a GenEval score of 82 for image generation alongside strong performance in text and vision understanding tasks. These results establish Uni-X as a parameter-efficient and scalable foundation for future unified multimodal modeling. Our code is available at `https://github.com/CURRENTF/Uni-X`.

## 1 Introduction

Vision-Language Models (VLMs) have demonstrated remarkable progress in multimodal understanding and reasoning, enabled by combining Large Language Models (LLMs) with powerful visual encoders (Liu et al., 2024c;b; Wang et al., 2024b; Team et al., 2024b). Motivated by this success, recent efforts to add image generation have led to **Unified Multimodal Models (UMMs)** (Team, 2025; Wang et al., 2024c; Wu et al., 2024b). However, many advanced UMMs rely on increasingly complex system designs to boost performance, including the addition of semantic image encoders (Wu et al., 2024a; Chen et al., 2025b; Deng et al., 2025; Wu et al., 2025a; Lin et al., 2025), the hybridization of autoregressive and diffusion paradigms (Wu et al., 2025a; Zhao et al., 2024; Ge et al., 2025; Deng et al., 2025; Xie et al., 2025; Zhou et al., 2024), or the introduction of task-specific branches and experts (Deng et al., 2025; Liao et al., 2025; Li et al., 2025d). While effective, this added complexity hinders scalability, limiting the degree of parameter sharing and reducing the potential for mutual benefits across tasks and modalities.

In contrast, **autoregressive (AR) UMMs** offer a simple yet powerful alternative. By treating visual inputs as a "foreign language" through vector quantization (VQ) (van den Oord et al., 2018; Esser et al., 2021b), they unify text and vision into a consistent token sequence, naturally extending the language-centric paradigm of LLMs (Wu et al., 2025c; Wang et al., 2024c). Despite this simplicity,

our experiments reveal a fundamental challenge: **fully modality-shared transformers trained jointly on multimodal inputs exhibit severe gradient conflicts.** Originally studied in multi-task learning (Yu et al., 2020; Shi et al., 2023), we are the first to transfer this concept to UMMs, uncovering inter-modality conflicts that hinder convergence and performance.

As illustrated in Figure 1, these conflicts are most pronounced in shallow (input) and deep (output) layers due to differing text-image statistics. In contrast, the middle layers, where representations become increasingly abstract and semantic (Meng et al., 2022; Geva et al., 2021b; Sun et al., 2025), show reduced conflicts and stronger cross-modal alignment. This suggests that an effective UMM should **respect modality-specific differences** rather than enforcing uniform parameter sharing across all layers.

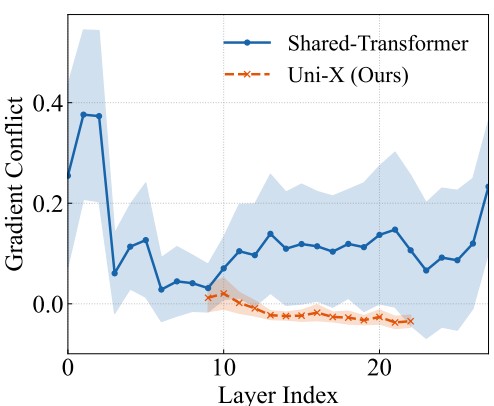

Figure 1: Gradient conflict analysis of down-projection weights in the FFN of a modality-shared transformer. The shared transformer exhibits severe conflicts in shallow and deep layers, with only partial mitigation in intermediate layers. In contrast, Uni-X avoids conflicts at both extremes and further alleviates them in the middle layers.

Guided by this observation, we introduce **Uni-X**, a *two-end-separated, middle-shared* architecture for unified multimodal modeling. In Uni-X, the shallow and deep layers are modality-specific, enabling specialized processing of distinct low-level distributions in text and vision, while the middle layers are shared to capture high-level semantic abstractions common to both. This **X-shaped architecture** not only mitigates the severe gradient conflicts at the two ends but also further alleviates residual conflicts in the shared middle layers by leveraging natural semantic alignment between modalities (Figure 1).

To demonstrate the effectiveness of Uni-X, we conduct extensive experiments under controlled training budgets and scaling regimes. Results show that Uni-X improves training efficiency and achieves stronger performance under identical conditions. Moreover, with larger data and model scales, our 3B-parameter Uni-X matches or surpasses the performance of existing 7B AR-based UMMs across both understanding and generation benchmarks, demonstrating its scalability and competitiveness. Our contributions are threefold:

- **Empirical Analysis:** We identify and quantify gradient conflicts between text and vision modalities in the shallow and deep layers of shared autoregressive transformers, attributing them to fundamental differences in their low-level statistical properties.
- **Model Design:** We propose Uni-X, a novel two-end-separated, middle-shared architecture that aligns model structure with modality characteristics by using modality-specific layers for low-level processing and a shared core for high-level semantic fusion.
- **Comprehensive Validation:** Extensive experiments demonstrate that Uni-X improves training efficiency and scales effectively, enabling a 3B model to achieve performance competitive with much larger 7B models across diverse multimodal benchmarks.

## 2 RELATED WORK

**Visual Language Models (VLMs).** The remarkable progress of LLMs (Touvron et al., 2023; Yang et al., 2024; Brown et al., 2020) has motivated researchers to extend them with visual cognition, giving rise to VLMs (Liu et al., 2024c; Achiam et al., 2023). Most VLMs leverage pre-trained visual encoders such as CLIP (Radford et al., 2021) or SigLIP2 (Tschannen et al., 2025) to extract semantic features from images, which are projected into the LLM's semantic space via multimodal adapters (Liu et al., 2024c;b; Beyer et al., 2024; Team et al., 2024a; Li et al., 2025a). This design enables strong multimodal understanding and reasoning but remains **asymmetric**: VLMs treat images only as inputs and cannot generate them, limiting synergy between perception and synthesis.

**Unified Multimodal Models (UMMs).** To enable such synergy, recent efforts have shifted toward UMMs, which aim to support both understanding and generation within a single framework (Team,

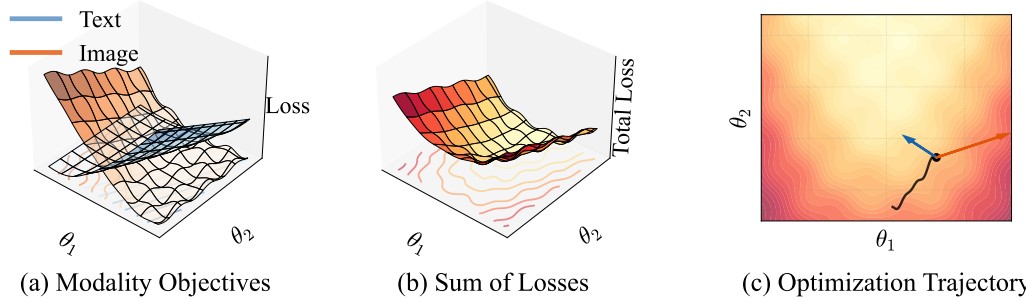

(a) Modality Objectives      (b) Sum of Losses      (c) Optimization Trajectory

Figure 2: Illustration of gradient conflict. (a) The loss landscapes of different modalities exhibit distinct geometries, creating potential conflicts in optimization direction. (b) The optimum of the sum of losses is different from the optimum of any single modality's loss. (c) In the presence of gradient conflict, the optimization trajectory becomes oscillating and suffers from slow convergence.

2025; Jin et al., 2024; Wu et al., 2024b). A natural extension is to adopt the autoregressive (AR) paradigm of LLMs by treating visual tokens as a "foreign language" via vector quantization (Wu et al., 2025c; Wang et al., 2024c). However, the distinct statistical properties of text and images often lead to **modality conflicts**, degrading performance in shared transformers (Team, 2025).

To mitigate this, several approaches increase architectural complexity: Mixture-of-Transformers (MoT) designs (Liao et al., 2025; Deng et al., 2025; Shi et al., 2025a) separate understanding and generation with distinct branches; Hybrid AR–diffusion frameworks (Zhao et al., 2024; Wu et al., 2025a; Dong et al., 2023; Ge et al., 2025) combine next-token prediction with diffusion-based image synthesis; and branching strategies such as UniFork (Li et al., 2025d) add task-specific deep heads. While effective on benchmarks, these methods sacrifice parameter sharing, complicate training, and weaken cross-modal benefits–the very goals UMMs were meant to unify.

**Comparison with Uni-X.** Uni-X builds on these insights but takes a different path. Instead of adding modules, Uni-X retains the simplicity of pure AR UMMs while mitigating modality conflict through a **two-end-separated, middle-shared** architecture: shallow and deep layers use modality-specific parameters to process low-level statistical differences, while intermediate layers are shared to exploit high-level semantic alignment. This X-shaped design avoids the rigidity of MoT or UniFork and the complexity of AR-diffusion hybrids, offering a lightweight yet effective solution. We further discuss other related works in Appendix A.1.

## 3 UNI-X

### 3.1 OBSERVATIONS

Before introducing the Uni-X architecture, we first analyze the gradient conflicts that arise when training modality-shared autoregressive transformers on multimodal data. We also provide an information-theoretic perspective to explain why such conflicts emerge.

**Definition of Gradient Conflict.** As illustrated in Figure 2, gradient conflict occurs when different optimization objectives induce gradients pointing in divergent directions, making joint optimization unstable and inefficient. To quantify gradient conflict in multimodal training, we use an early checkpoint of a fully shared transformer. From this model checkpoint, we compute the *average gradients* for specific parameter groups (e.g., FFN down-projection weights) using over 60 mini-batches and totaling 2M tokens, to ensure obtaining stable gradients.

Specifically, we first compute the average text gradient, $g_{\texttt{text}}$. This is obtained by exclusively performing forward and backward passes on $D_{\texttt{text}}$, a text-only subset filtered from the pre-training data. Next, we compute the average image-text gradient, $g_{\texttt{img}}$, using an analogous subset $D_{\texttt{img}}$ containing image-text pairs. The raw inter-modal similarity is then measured as the cosine similarity between these two average gradients:

$$S_{\texttt{inter}} = \cos(g_{\texttt{text}}, g_{\texttt{img}}). \tag{1}$$

However, since transformer layers have distinct roles at depth (Sun et al., 2025; Geva et al., 2021a), the resulting raw similarity $S_{\texttt{inter}}$ is biased and cannot be directly compared. To correct for this, we estimate a baseline similarity $S_{\texttt{base}}$ that reflects the model's intrinsic gradient consistency on a unified data distribution. We randomly shuffle the full multimodal dataset $D_{\texttt{all}}$ and split it into two disjoint halves, $D_{\texttt{any}}^1$ and $D_{\texttt{any}}^2$. Their respective average gradients, $\boldsymbol{g}_{\texttt{any}}^1$ and $\boldsymbol{g}_{\texttt{any}}^2$, yield:

$$S_{\texttt{base}} = \cos(\boldsymbol{g}_{\texttt{any}}^1, \boldsymbol{g}_{\texttt{any}}^2). \tag{2}$$

This value represents the expected gradient similarity when gradients originate from the same underlying distribution. Thus, we define the **gradient conflict** $c_g$ as the deviation from this baseline:

$$c_g = -(S_{\texttt{inter}} - S_{\texttt{base}}). \tag{3}$$

A high $S_{\texttt{base}}$ indicates the model's gradients are stable, whereas a much lower $S_{\texttt{inter}}$ suggests that the text-only and image-text data push the shared model parameters in conflicting directions, resulting in a large positive $c_g$. This provides a principled, layer-wise measure of inter-modal *disagreement* and reveals where and why conflicts are most severe.

**Empirical Findings.** Figure 1 shows gradient conflict profiles ($c_g$) across depth. In modality-shared transformers, conflicts are most pronounced in shallow layers (near input) and deep layers (near output), while intermediate layers exhibit weaker conflicts. Experiments further reveal that Uni-X avoids conflicts at both extremes and reduces residual conflicts in the middle, validating its structural design. Additional analyses of other modules and the relationship between gradient conflict and data, as well as its impact on model performance, are provided in Appendix A.5.

**Why Do Conflicts Arise? Vision as a "Foreign Language"** To explain these observations, we examine whether vision behaves like a "foreign language" when tokenized. Using the VQ tokenizer (Team, 2025), images are represented as discrete token sequences, formally similar to text. Then, we define conditional entropy based on $n$-gram. When $n = 1$, the calculation reduces to ordinary information entropy. For $n > 1$, the conditional entropy is computed as follows:

$$H_n = -\sum p(w_n \mid w_1, w_2, \cdots, w_{n-1}) \log p(w_n \mid w_1, w_2, \cdots, w_{n-1}). \tag{4}$$

Results (Figure 3) show that image tokens exhibit far higher entropy than natural languages such as English, German, or Chinese. While languages differ in grammar and lexicon, their token statistics remain closer to each other than to images. This means visual sequences are inherently harder to predict, requiring modeling of long-range, spatially entangled dependencies.

As information theory suggests, sequences with higher (conditional) entropy are inherently harder to predict, requiring models to learn longer-range dependencies and more complex patterns. Thus, when a shared transformer jointly processes low-entropy, grammatical text with high-entropy, spatially complex vision, shallow and deep layers are forced to reconcile conflicting low-level distributions, producing strong gradient conflicts. In contrast, inter-

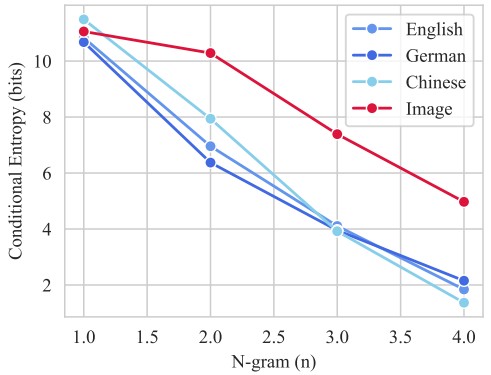

Figure 3: Conditional entropy of images and natural languages. Image token sequences encoded by the VQ tokenizer exhibit substantially higher entropy, indicating greater difficulty in prediction.

mediate layers, where representations become more abstract and semantic, naturally align across modalities, explaining the reduced conflicts observed in practice.

## 3.2 MODEL ARCHITECTURE

Motivated by these findings, we propose **Uni-X**, an architecture designed to explicitly align model structure with modality characteristics.

**Core Principle.** As illustrated in Figure 4, Uni-X follows a two-end-separated, middle-shared design. The shallow and deep layers are duplicated into parallel modality-specific branches, ensuring

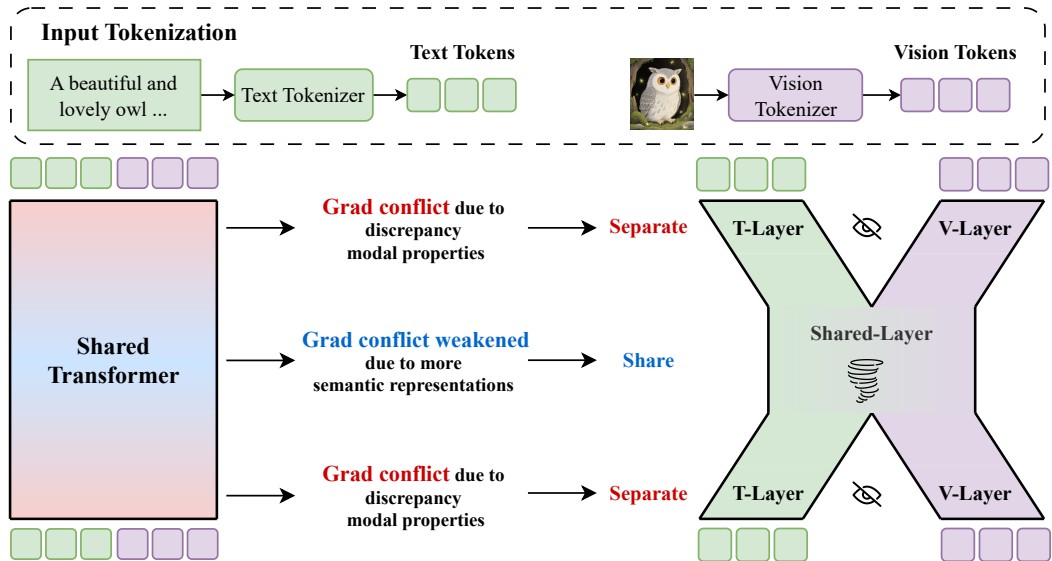

Figure 4: Illustration of the proposed **Uni-X** architecture compared with a standard modality-shared transformer. The baseline shared transformer (**left**) encounters gradient conflicts in shallow and deep layers due to the mismatched statistical properties of vision and text tokens. In contrast, Uni-X (**right**) adopts a two-end-separated, middle-shared design: modality-specific layers at both ends handle low-level feature processing, while a shared central block performs high-level semantic fusion. This structure aligns the architecture with the inherent characteristics of each modality and effectively mitigates gradient conflicts.

independent handling of text and vision during early feature extraction and final token projection. The intermediate layers remain shared, enabling high-level semantic fusion across modalities. This X-shaped separation-and-sharing balances modality specialization with semantic alignment.

**Input Tokenization.** For visual inputs, we employ the VQGAN tokenizer (Esser et al., 2021a) from Chameleon (Team, 2025) to encode $512 \times 512$ images into a $32 \times 32$ grid of visual tokens from an 8,192-entry codebook. To accommodate these new tokens, we expand the vocabulary and corresponding embedding matrix of the base LLM. Textual inputs are processed via the standard BPE tokenizer. The final token sequence is structured as `<BOI>[Image]<EOI>[Text]<BOS>` for image understanding tasks and `[Text]<BOI>[Image]<EOI><BOS>` for image generation tasks. This unified tokenization enables AR training across modalities. While Uni-X can handle interleaved multimodal sequences, this study focuses on non-interleaved inputs.

**Forward Propagation.** Given a pre-trained LLM with $L$ layers, denoted as $\{\texttt{Layer}_\texttt{t}^i\}_{i=0}^{L-1}$, we partition them into three sections: The initial $N$ layers and the final $M$ layers constitute the "separated layers," while the intermediate layers form the "shared layers." Within the separated blocks, we introduce a new set of vision-specific layers, $\{\texttt{Layer}_\texttt{v}^i\}$, which operate in parallel with the original text layers, $\{\texttt{Layer}_\texttt{t}^i\}$.

To manage the data flow, we introduce a binary mask $\boldsymbol{M}_\texttt{v} \in \{0,1\}^n$ to identify the positions of visual tokens. At any given layer $l$, the complete hidden states $\boldsymbol{H}^l$ can be partitioned into text-specific states $\boldsymbol{H}_\texttt{t}^l = \boldsymbol{H}^l[\sim \boldsymbol{M}_\texttt{v}]$ and vision-specific states $\boldsymbol{H}_\texttt{v}^l = \boldsymbol{H}^l[\boldsymbol{M}_\texttt{v}]$.

The forward propagation in Uni-X is defined as follows:

$$\boldsymbol{H}_x^{l+1} = \begin{cases} \texttt{Layer}_x^l(\boldsymbol{H}_x^l) & \text{if } l < N \text{ or } l \geq L - M, \\ \left[\texttt{Layer}_\texttt{t}^l(\boldsymbol{H}^l)\right]_x & \text{otherwise,} \end{cases} \quad (5)$$

where $x \in \{\texttt{t}, \texttt{v}\}$ denotes the modality (text or vision). In the "otherwise" case, $[\cdot]_x$ indicates selecting the subset of the output hidden states corresponding to modality $x$.

Importantly, unlike other architectures (Li et al., 2025d; Deng et al., 2025; Shi et al., 2025a), the vision and text modalities remain strictly isolated within the separated blocks, with no cross-modal

interaction. This forces the model to learn robust unimodal representations before they are fused in the shared block and after they are separated for modality-specific output generation.

**Training Objective.** Following the standard paradigm for AR models, Uni-X is trained to predict the next token in a sequence containing both text and visual tokens. The training objective is to minimize the cross-entropy loss over the vocabulary for each token. The loss function $\mathcal{L}$ for a given sequence $\mathcal{S} = (s_1, s_2, \cdots, s_T)$ is defined as:

$$\mathcal{L} = -\sum_{i=1}^{T} \log P(s_i \mid s_{<i}).$$  (6)

This simple yet effective objective enables the model to learn both understanding and generation capabilities across modalities within a single, unified framework.

**Design Rationale.** Unlike prior architectures that rely on auxiliary semantic encoders (Wu et al., 2024a; Chen et al., 2025b), hybrid AR–diffusion pipelines (Wu et al., 2025a; Zhao et al., 2024), or task-specific branching structures such as MoT (Liao et al., 2025) and UniFork (Li et al., 2025d), Uni-X maintains the simplicity of a pure autoregressive framework. Its two-end-separated, middle-shared structure is motivated directly by empirical evidence of gradient conflicts, aligning the model design with the statistical characteristics of each modality. By isolating low-level modality-specific processing while preserving a shared semantic core, Uni-X avoids the complexity and overhead of multi-expert or dual-paradigm systems, yet achieves competitive or superior performance. This balance of architectural simplicity, empirical grounding, and scalability makes Uni-X a practical foundation for unified multimodal modeling.

## 4 EXPERIMENTS

We evaluate Uni-X from two complementary perspectives: (1) **Efficiency under identical training conditions**, where Uni-X and baseline architectures are trained on the same data and resources, enabling fair comparisons of efficiency and performance. (2) **Scaling within resource constraints**, where we maximize dataset size and training duration to examine Uni-X's scalability and competitiveness against larger state-of-the-art models.

### 4.1 EXPERIMENTAL SETUP

**Pre-training Datasets.** Our pre-training stage was designed to build a strong foundation in both language and vision. To preserve general text generation capabilities, we utilized a diverse set of text corpora: the high-quality Chinese dataset CCI3-H (Wang et al., 2024a), English datasets DCLM (Li et al., 2025c) and Fineweb-Edu (Penedo et al., 2024), and the StarcoderData (Li et al., 2023b) corpus, as integrating code is known to boost general model performance (MA et al., 2024). For multimodal pre-training, we used public benchmarks like ImageNet (Russakovsky et al., 2015) and JourneyDB (Pan et al., 2023), complemented by a substantial internally collected dataset of 40 million images, which were captioned using the powerful Intern-VL model Chen et al. (2024). Following the methodology of Liquid (Wu et al., 2025c), we diversified our training data by randomly reversing 20% of the text-to-image pairs to serve as image-captioning tasks. The final pre-training data consists of 72B text tokens and 65B vision tokens.

**Supervised Fine-Tuning (SFT).** We further refined the model with 3B SFT tokens. For vision understanding, we employed MiniGemini (Li et al., 2024a) and FineVision (HuggingFaceM4, 2025). To improve text understanding and general instruction-following, we utilized OpenOrca (Mukherjee et al., 2023). Additionally, to refine the quality of image generation, we leveraged Blip3o-60k (Chen et al., 2025a) and ShareGPT4o (Chen et al., 2023).

**Benchmarks.** Evaluation covered text-only, image generation, and multimodal understanding tasks. For text-only tasks, we employed ARC-Easy/Challenge (**ARC-E/ARC-C**) (Clark et al., 2018), Wino-Grande (**WinoG**) (Sakaguchi et al., 2020), **BoolQ** (Clark et al., 2019), and **MMLU** (Hendrycks et al., 2021). For image generation, we used **GenEval** (Ghosh et al., 2023) and DPG-Bench (**DPG**) (Hu et al., 2024). For the GenEval benchmark, we followed Bagel (Deng et al., 2025) and employed an LLM to rewrite shorter prompts into more detailed ones to better assess instruction following. For multimodal understanding, we used SEEDBench (**SEED**) (Li et al., 2023a), **MME** (Fu et al., 2024), **POPE** (Li et al., 2023c), and MMBench (**MMB**) (Liu et al., 2024d).

Table 1: The text performance of Uni-X compared to other models.

| Model | # Params. | ARC-E | ARC-C | WinoG | BoolQ | MMLU | Avg. ↑ |
|---|---|---|---|---|---|---|---|
| **Janus-Pro** (Chen et al., 2025b) | 7B | 70.4 | 40.9 | 66.1 | 80.2 | 49.3 | 61.4 |
| **VILA-U** (Wu et al., 2024b) | 7B | 51.6 | 34.0 | 57.3 | 70.6 | 25.5 | 47.8 |
| **Chameleon** (Team, 2025) | 7B | 76.1 | 46.5 | 70.4 | 81.4 | 52.1 | 65.3 |
| **Liquid** (Wu et al., 2025c) | 7B | 75.6 | 49.0 | 72.7 | 81.0 | 56.0 | 66.9 |
| **Uni-X** | 3B / 4.5B | 79.0 | 47.9 | 68.9 | 82.2 | 57.6 | 67.1 |

Table 2: The image generation and multimodal understanding performance of Uni-X compared to other models. In the **# Params** column $x/y$, $x$ and $y$ represent the number of active parameters and the total parameters, respectively. $\dagger$ represents the model variant that performs semantic alignment. $\ddagger$ represents the rewriting of the prompt during evaluation. $\heartsuit$ indicates that it has been trained on more image-text data.

| Model | # Tokens | # Params. | GenEval | DPG | MME | POPE | MMB | SEED |
|---|---|---|---|---|---|---|---|---|
| *Autoregressive meets Diffusion* | | | | | | | | |
| **Bagel** (Deng et al., 2025) | 5.1T | 7B / 14B | 88[‡] | 85.0 | - | - | 85.0 | - |
| **Bilp3o** (Chen et al., 2025a) | - | 4B / 9B | 81 | 79.3 | 1,527.7 | - | 78.6 | 73.8 |
| **X-Omni** (Geng et al., 2025) | ~1T | 10B / 20B | 83[‡] | 87.6 | - | 89.3 | 74.8 | 74.1 |
| **Show-o** (Xie et al., 2024) | ~500B | 1.3B | 68 | - | 1,097.2 | 80.0 | - | - |
| **Show-o**[†] (Xie et al., 2024) | ~500B | 1.3B | 69 | - | 1,232.9 | 84.5 | - | - |
| *Autoregressive w/ Semantic Encoder* | | | | | | | | |
| **NextStep1** (Team et al., 2025) | ~1T | 14B | 73[‡] | 85.2 | - | - | - | - |
| **Janus-Pro** (Chen et al., 2025b) | ~300B | 7B | 80 | 84.1 | - | 87.4 | 79.2 | 72.1 |
| **VILA-U** (Wu et al., 2024b) | - | 7B | - | - | 1,336.2 | 83.9 | - | 56.3 |
| **Liquid**[†] (Wu et al., 2025c) | - | 8B | - | - | 1,448.0 | 83.2 | - | - |
| *Autoregressive w/o Semantic Encoder* | | | | | | | | |
| **Chameleon** (Team, 2025) | 9.2T | 34B | 39 | - | 604.5 | - | 32.7 | - |
| **LWM** (Liu et al., 2024a) | ~500B | 7B | 47 | - | - | 75.2 | - | - |
| **EMU3** (Wang et al., 2024c) | - | 8B | 66[‡] | 80.6 | 1,243.8 | 85.2 | 58.5 | 68.2 |
| **Liquid** (Wu et al., 2025c) | ~90B | 7B | 68[‡] | 79.8 | 1,107.2 | 81.1 | - | - |
| **Uni-X** | 140B | 3B / 4.5B | 82[‡] | 79.8 | 1,158.3 | 83.6 | 59.3 | 60.2 |
| **Uni-X**[♡] | 240B | 3B / 4.5B | 83[‡] | 80.3 | 1,228.2 | 84.6 | 62.7 | 59.8 |

**Implementation Details.** We conducted ablation studies on Qwen2.5-1.5B (Yang et al., 2024) and scaled to Qwen2.5-3B. We used the VQGAN tokenizer (Esser et al., 2021a) from Chameleon (Team, 2025) to encode $512 \times 512$ images into $32 \times 32$ discrete tokens. Our codebase is built upon the Liquid (Wu et al., 2025c) and HuggingFace Transformers (Wolf et al., 2019) libraries. Training was accelerated using Flash Attention 2 (Dao et al., 2022) and DeepSpeed ZeRO2 (Aminabadi et al., 2022). When generating images, we uniformly set the classifier-free guidance (CFG) to 4.0.

## 4.2 RESULTS AND ANALYSIS

**Scaling Experiment.** In this experiment, we aim to demonstrate that Uni-X can scale effectively and is not limited to small-scale training data. We expand the dataset size to 140B total tokens for Uni-X, using Qwen2.5-3B as the base model, and extend the training duration to achieve improved performance. The evaluation is conducted against SOTA models, some of which have been trained on trillions of tokens. As presented in Table 1, Uni-X robustly maintains the strong language capabilities of its base model. With an average score of 67.1 across five text benchmarks, our 3B Uni-X model outperforms several larger 7B models. This demonstrates that our design successfully mitigates modality conflict without sacrificing performance on fundamental language understanding tasks.

For image generation, as detailed in Table 2, Uni-X achieves a strong score of 82 on GenEval, a result that surpasses many models with more parameters and underscores the effectiveness of

Table 3: Image edit results on ImgEdit-Bench. $\heartsuit$ denotes models trained on extra image-text data.

| Model | # Params. | Add | Adjust | Extract | Replace | Remove | Background | Style | Hybrid | Action | Overall↑ |
|---|---|---|---|---|---|---|---|---|---|---|---|
| GPT-4o | - | 4.61 | 4.33 | 2.90 | 4.35 | 3.66 | 4.57 | 4.93 | 3.96 | 4.89 | 4.20 |
| ICEdit | 12B | 3.58 | 3.39 | 1.73 | 3.15 | 2.93 | 3.08 | 3.84 | 2.04 | 3.68 | 3.05 |
| AnyEdit | 4B | 3.18 | 2.95 | 1.88 | 2.47 | 2.23 | 2.24 | 2.85 | 1.56 | 2.65 | 2.45 |
| UltraEdit | 4B | 3.44 | 2.81 | 2.13 | 2.96 | 1.45 | 2.83 | 3.76 | 1.91 | 2.98 | 2.70 |
| Step1X-Edit | 12B | 3.88 | 3.14 | 1.76 | 3.40 | 2.41 | 3.16 | 4.63 | 2.64 | 2.52 | 3.06 |
| Bagel | 7B / 14B | 3.56 | 3.31 | 1.70 | 3.30 | 2.62 | 3.24 | 4.49 | 2.38 | 4.17 | 3.20 |
| Uni-X$^\heartsuit$ | 3B / 4.5B | 3.57 | 3.18 | 2.06 | 3.94 | 3.82 | 3.38 | 4.21 | 3.16 | 3.63 | 3.44 |

Table 4: Performance and training efficiency comparison of different model architectures under identical training conditions. Training efficiency is measured by the number of tokens processed per second per GPU. ♠ indicates that the baseline has been adapted for our experimental setting (see Appendix A.3 for specific details); $\diamond$ represents calculating the loss on the instruction part during the training of image-text data.

| Model | #Params. | MMLU | GenEval | MMB | Avg. ↑ | Efficiency ↑ |
|---|---|---|---|---|---|---|
| Shared Transformer | 1.5B | 50.0 | 33.6 | 30.3 | 38.0 | 16,380 |
| MoT♠ Deng et al. (2025) | 1.5B / 3B | 48.0 | 26.0 | 30.0 | 34.6 | 12,658 |
| HardMoE | 1.5B / 2.3B | 50.3 | 42.8 | 30.7 | 41.3 | 14,657 |
| UniFork♠ (Li et al., 2025d) | 1.5B / 2.3B | 50.1 | 12.4 | 25.9 | 29.5 | 15,481 |
| Uni-X (9:5)$^\diamond$ | 1.5B / 2.3B | 48.5 | 34.8 | 29.8 | 37.7 | 15,642 |
| Uni-X (9:5) | 1.5B / 2.3B | 50.1 | 43.3 | 31.5 | 41.6 | 15,595 |

our architecture in producing high-quality images. We also tested T2I-CompBench (Huang et al., 2025) and MSCOCO (Lin et al., 2015), as shown in Appendix A.4. Uni-X similarly exhibited strong performance with fewer parameters. Regarding vision understanding (Table 2), while Uni-X's scores are slightly lower than some state-of-the-art models, we observe a clear trend: models that incorporate an additional semantic image encoder, such as Janus-Pro (Chen et al., 2025b) and the semantically aligned variants of Liquid (Wu et al., 2025c) and Show-o (Xie et al., 2024), tend to achieve substantially higher performance on understanding benchmarks like MMBench and SEED.

In contrast, among models that do not rely on a separate semantic encoder, Uni-X's performance is commendable and holds its ground against strong competitors like EMU3 (Wang et al., 2024c). This suggests that our architecture effectively harnesses the inherent capabilities of the autoregressive framework for vision understanding. We speculate that the relatively weaker understanding performance might be partially caused by the insufficient utilization of the VQ tokenizer's codebook. We analyzed the token sequences encoded from 1 million images and found that, although there are 8,192 tokens available in the tokenizer, only $\approx$3,127 are being utilized. Meanwhile, EMU3 uses 4096 tokens to represent a $512 \times 512$ image, which provides more fine-grained information. However, this $4\times$ token count severely impacts its image generation speed, as shown in Appendix A.6.

For image editing, we conducted tests on ImgEdit (Ye et al., 2025) as shown in Table 3. Uni-X achieved better results than Bagel, even with less training data and fewer parameters than Bagel. This demonstrates that the high-level semantic unification of Uni-X enhances its image editing capabilities.

**Identical Training Conditions.** To validate the effectiveness of Uni-X, we conducted ablation experiments on a smaller dataset and a slightly reduced base model Qwen2.5-1.5B, due to resource constraints. To ensure consistency in performance comparisons, we limited the dataset to 28B tokens, of which 13.7B are vision tokens. The experiments were conducted using a learning rate (LR) of $5 \times 10^{-5}$, warmup ratio 0.03, and constant LR scheduler, with batch size 17,560 tokens per GPU.

The selected baselines include: (1) **Shared Transformer**, which continues multimodal pre-training based on Qwen2.5-1.5B; (2) **Mixture-of-Transformers (MoT)** (Deng et al., 2025), where prior work replicates an additional transformer to handle image generation tasks, while the original LLM backbone focuses on text-only and image understanding tasks. Under our experimental setup, vision tokens are allocated to the duplicated transformer; (3) **Hard-Route MoE (HardMoE)**, which

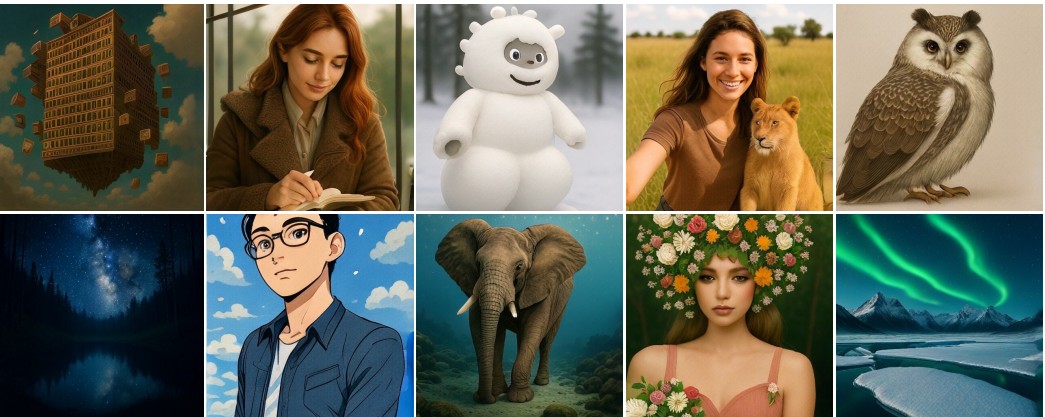

Figure 5: Qualitative examples of Uni-X image generation. The results highlight its ability to produce diverse, high-quality visuals that follow prompts with both creativity and fine-grained detail.

introduces a vision expert specifically for the vision modality, assigning vision tokens to this expert for computation guided by the vision mask; and (4) **UniFork** (Li et al., 2025d), which creates a task-specific deep branch for image generation. For all the baselines, we ignored the instruction during training to enhance cross-modal performance and ensure a fair comparison.

Results (Table 4) show that Uni-X achieves the best overall performance under consistent training conditions. Specifically, our Uni-X (9:5) configuration attains an average score of 41.6, significantly outperforming the standard baselines. While HardMoE is competitive, achieving a score of 41.3, Uni-X still holds a slight advantage. Moreover, HardMoE and UniFork are orthogonal and can be combined. In terms of training efficiency, although the baseline shared transformer is the fastest due to having the fewest parameters, Uni-X achieves a high throughput, which is considerably more efficient than the less performant MoT architecture. These findings confirm that Uni-X's design offers a more effective trade-off between performance and computational efficiency.

It is worth noting that the architectures of MoT and UniFork have been adapted to fit our VQ+AR setup to avoid discrepancies in efficiency between paradigms such as diffusion and AR+diffusion. Specific details can be found in Appendix A.3. A comparison of training efficiency across paradigms lies beyond the scope of this work and will be considered in future research.

**Case Study.** In Figure 5, we present a curated selection of images generated by Uni-X to qualitatively assess its capabilities. Despite the relatively limited number of tokens used during training, the model demonstrates a strong ability to produce clear, aesthetically pleasing images that exhibit robust instruction-following capabilities. The examples showcase Uni-X's versatility in handling a wide range of creative and complex prompts. For instance, the model can generate imaginative fantasy scenes, such as a gigantic library floating above the clouds, and surreal compositions, like a realistic elephant walking on the ocean floor. Furthermore, Uni-X success-

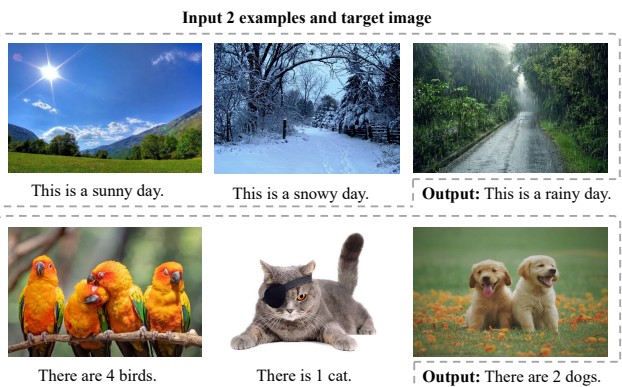

Figure 6: Demonstration of Uni-X's in-context learning. The model follows few-shot examples to perform tasks such as image description (1st line) and object counting (2nd line).

fully adheres to specific artistic style requests, as seen in the detailed anime-style portrait, and renders fine details with high fidelity, exemplified by the intricate feather patterns of the owl. These case studies verify that Uni-X can effectively translate complex textual descriptions into high-quality visual outputs. The specific prompts used for these generations are provided in Appendix A.2.

**In-Context Learning.** Although Uni-X was not explicitly trained on interleaved multimodal data, we conducted an evaluation to assess its emergent in-context learning (ICL) capabilities. As illustrated in Figure 6, the model was presented with few-shot examples, where several image-text pairs were provided as context before a final query image was presented without its corresponding description.

The results demonstrate that Uni-X can successfully interpret the contextual examples and apply the learned pattern to the target image. For instance, in the top row of Figure 6, the model correctly identifies the weather in the target image as a "rainy day," adhering to the simple descriptive format ("This is a... day.") established by the preceding examples. Also, Uni-X exhibits the ability to perform more reasoning tasks, such as object counting. This suggests that the model is not merely mimicking sentence structure but is performing cross-modal reasoning at a semantic level.

**Ignore Instruction in Training.** Ignoring the loss of the instruction part during training is a common technique in supervised fine-tuning. However, its role in pretraining is rarely emphasized. Following Liquid (Wu et al., 2025c), we applied the same "ignore instruction" strategy during pretraining.

Specifically, no loss mask was applied for pure text data. For text-image pairs, in text-to-image tasks, the loss calculation excluded the text instruction tokens. Similarly, for image captioning tasks, the loss corresponding to the image tokens was masked. As demonstrated in our experimental results, Table 4, this approach significantly enhanced the model's capability to generate images.

We believe there might be several reasons for this: 1) This mask forces the model to learn the relationship between the two modalities rather than relying on the prior distribution of images, thereby enhancing its instruction-following capability. 2) It serves as a form of loss regularization. For text-image pair data, the number of image tokens is fixed at 1024, while the average number of text tokens is around 120. By masking, we ensure that the gradient magnitude generated by the loss is only dependent on the reverse ratio we set.

Table 5: Performance comparison of different Uni-X configurations. Here, $x : y$ denotes the number of shallow separated layers $x$ and deep separated layers $y$, respectively. The total number of layers is $n = 28$. The split points are $x$ and $n - y$, respectively.

| Configuration | MMLU | GenEval | MMB | Avg. ↑ |
|---|---|---|---|---|
| **Uni-X** (3:3) | 48.7 | 37.3 | 30.7 | 38.9 |
| **Uni-X** (7:7) | 49.6 | 41.3 | 29.4 | 40.1 |
| **Uni-X** (11:11) | 49.7 | 37.5 | 32.1 | 39.8 |
| **Uni-X** (3:11) | 50.0 | 32.9 | 31.0 | 38.0 |
| **Uni-X** (5:9) | 50.1 | 39.2 | 28.0 | 39.1 |
| **Uni-X** (9:5) | 50.1 | 43.3 | 31.5 | 41.6 |
| **Uni-X** (11:3) | 49.8 | 25.1 | 31.9 | 35.6 |

**Number of Separated Layers.** We investigate how the number and distribution of separated layers affect performance (Table 5). Varying the total number of separated layers produces an $n$-shaped trend: more separation improves modality-specific low-level processing, but too many layers reduce shared middle layers, weakening semantic fusion and cross-modal reasoning. The best overall performance is achieved with 14 separated layers.

We then examine shallow-deep ratios under this setting. A 9:5 split (slightly more shallow than deep layers) performs best, indicating that early processing of low-level features, where text and vision differ most, benefits more from modality-specific capacity than the final generation stage. These results provide strong empirical support for the Uni-X design. We also explored text layers and vision layers with different numbers of separate layers, and the results are shown in Appendix A.7.

## 5 CONCLUSIONS

In this work, we identified gradient conflicts as a fundamental limitation of shared AR UMMs, particularly in the shallow and deep layers where vision and text exhibit highly divergent low-level statistics. To address this challenge, we proposed Uni-X, a two-end-separated, middle-shared architecture that explicitly aligns model structure with modality characteristics. By isolating low-level processing into modality-specific branches while maintaining a shared semantic core for high-level fusion, Uni-X effectively mitigates inter-modal conflicts without adding architectural complexity. Extensive experiments show that this X-shaped design allows a 3B-parameter Uni-X model to deliver performance competitive with much larger 7B UMMs across diverse multimodal benchmarks. These findings establish Uni-X as both a scalable and parameter-efficient foundation, paving the way for future research in unified multimodal modeling.

ACKNOWLEDGMENTS

We would like to thank the anonymous reviewers for their insightful feedback and constructive suggestions, which have significantly improved the quality of this paper. This work was supported by the National Natural Science Foundation of China (NSFC) under Grant Nos. 62125201, U24B20174, and U25B6003.

ETHICS STATEMENT

This research aims to advance the field of artificial intelligence, particularly in the area of Unified Multimodal Models. We recognize that, like other powerful generative models, the technologies proposed in this study also carry potential risks of misuse, such as the creation of misinformation, biased, or harmful content. Our primary objective is to explore architectural efficiency to build more powerful and scalable models, and we believe this will make a valuable contribution to science.

The datasets used to train and fine-tune our models are primarily publicly available and widely used benchmark datasets in the academic community. For any internally collected data, we have ensured that its acquisition and processing adhere to principles of responsibility. We have not specifically filtered web-based datasets for bias, and therefore, the model may reflect social biases present in the data. We encourage responsible downstream use and further research into mitigating the potential negative impacts of generative models. Our work is intended solely for research purposes and is shared with the community to foster innovation and deepen understanding.

REPRODUCIBILITY STATEMENT

We are committed to ensuring the full reproducibility of our work. To facilitate verification and future extensions, we provide detailed descriptions of the model architecture, training setup, datasets, evaluation protocols, and implementation details throughout the main paper and appendix.

**Code.** The complete source code for the Uni-X architecture, training pipeline, and evaluation scripts is publicly available at: `https://github.com/CURRENTF/Uni-X`. The repository includes model definitions, data preprocessing utilities, configuration files, and instructions for reproducing both ablation and scaling experiments.

**Architecture and Implementation.** The description of the Uni-X architecture is provided in Section 3.2. Implementation details, including the base models (Qwen2.5-1.5B and Qwen2.5-3B), the VQGAN tokenizer configuration, training framework (Liquid and HuggingFace Transformers), optimization tools (FlashAttention 2 and DeepSpeed ZeRO2), and image generation settings (e.g., CFG = 4.0), are documented in Section 4.1. Adaptations made to baseline methods under the VQ+AR setting are fully described in Appendix A.3 to ensure fair and transparent comparison.

**Datasets and Evaluation Benchmarks.** All pre-training datasets (including text and multimodal corpora) and supervised fine-tuning datasets are listed in Section 4.1, along with token statistics. Evaluation benchmarks for text-only tasks, image generation, multimodal understanding, and image editing are also described in Section 4.1 and corresponding result tables. Where prompt rewriting or evaluation-specific adjustments were applied (e.g., GenEval prompt expansion), these procedures are explicitly documented.

**Hyperparameters and Training Settings.** Key training hyperparameters for ablation experiments, including learning rate, warmup ratio, batch size, and scheduler details, are specified in Section 4.1. Scaling experiments and total token counts (e.g., 140B / 240B settings) are also clearly reported in Section 4.2. These details ensure that experimental comparisons are conducted under fully specified and reproducible conditions.

We believe that the combination of open-source code, precise architectural definitions, transparent dataset descriptions, and explicitly reported training hyperparameters enables the research community to faithfully reproduce our results and build upon the Uni-X framework for future unified multimodal modeling research.

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

## A   APPENDIX

### A.1   OTHER RELATED WORK

**Other UMMs** Recent advancements in Unified Multimodal Models (UMMs) have explored various architectural and methodological innovations. For instance, OneCAT (Li et al., 2025b) proposes a decoder-only autoregressive architecture that utilizes modality-specific Mixture-of-Experts (MoE) structures to achieve efficient multimodal fusion without visual encoders. VARGPT (Zhuang et al., 2025) unifies visual understanding and generation through a seamless combination of next-token and next-scale prediction paradigms. To harmonize multi-granularity representations, the Harmon framework (Wu et al., 2025d) leverages a shared masked autoregressive encoder. Furthermore, MetaQueries (Pan et al., 2025) introduces learnable queries to transfer reasoning capabilities from frozen MLLMs to diffusion decoders.

**UMMs for Efficient Agent** The capabilities of vision-language models are continuously evolving, driving their application in specialized vertical domains and complex agentic tasks. For example, the emerging "Deep Research" paradigm (Shi et al., 2025b), iterative self-incentivization frameworks (Shi et al., 2025c), and domain-specific benchmarks like DeepShop (Lyu et al., 2025b) highlight the critical importance of tool utilization and autonomous multi-step interactions in real-world scenarios. Currently, translating these capabilities into complex tasks often relies on cumbersome multi-agent or multi-model pipelines. Instances of this include cooperative multi-agent systems for entity recognition (Wang et al., 2025) and alignment optimization (Lyu et al., 2025a), as well as specialized fragmented models for fine-grained document grounding (Zhou et al., 2025), search and recommendation (Zhao et al., 2026), and preference optimization (Wu et al., 2025b). However, these multi-model pipelines inevitably generate extremely long contexts and heavy computational burdens. While these efficiency issues can be partially alleviated by inference acceleration and efficient training methods like OmniKV (Hao et al., 2025b), SnapKV (Li et al., 2024b), and Low-Rank Clone (Hao et al., 2025a), relying on fragmented systems lacks architectural elegance (Shen et al., 2026; Hao et al., 2026). In contrast, UMMs offer a more concise and efficient alternative, enabling a single unified model to seamlessly handle diverse reasoning, tool-calling, and multimodal tasks without the cumbersome overhead of complex multi-agent interactions.

### A.2   PROMPTS OF IMAGE GENERATION

Table 6 lists the prompts corresponding to the generated images shown in Figure 5. The prompts are presented in the same order as the images: left to right, top to bottom. These examples highlight the diversity of tasks, ranging from descriptive captions to creative scene generation.

### A.3   BASELINE IMPLEMENTATION DETAILS

To ensure fair comparisons, we adapt baseline methods to the VQ+AR setting used in our study. For Mixture-of-Transformers (MoT) (Deng et al., 2025; Shi et al., 2025a; Liao et al., 2025), the

Table 6: Prompts used for the image generation examples shown in Figure 5.

| No. | Prompt |
|---|---|
| 1 | A gigantic library floats above the clouds, its appearance resembling a suspended castle. Every book emits a faint glow and drifts through the air with the gentle breeze. |
| 2 | A highly realistic close-up photo featuring a beautiful 35-year-old red-haired woman, writing in her diary on her balcony. She is dressed in warm yet stylish clothing. |
| 3 | A happy snowman. |
| 4 | A woman and her little lion taking a selfie on the grassland. |
| 5 | A beautiful owl with sleek feathers and lively eyes, its round head adorned with two furry ears. The elegant pattern is formed by the interweaving of snow-white down and deep brown flight feathers, making it appear both stunning and endearing. |
| 6 | A clearing in a deep, mysterious forest, with a mirror-like pond at its center, the water reflecting a night sky filled with the Milky Way. |
| 7 | A handsome 24-year-old boy stands in the center, with a sky-colored background. He is wearing glasses, and the art style is very detailed, in anime style. |
| 8 | A realistic photo of an elephant walking on the ocean floor. |
| 9 | An elegant and charming lady whose hair is entirely made up of blooming flowers, resembling a masterpiece of nature. The flowers are of various types, possibly including delicate roses, fresh daisies, vibrant sunflowers, or other colorful blossoms. |
| 10 | A magnificent landscape photo depicting the northern lights dancing above the snow-capped mountain ranges in Iceland. |

Table 7: The T2I-CompBench and MSCOCO performance of Uni-X. $^\heartsuit$ indicates that it has been trained on more image-text data.

| Model | # Params. | T2I-Color | T2I-Shape | T2I-Texture | T2I-Avg. ↑ | MSCOCO CLIP-T |
|---|---|---|---|---|---|---|
| SDXL | 3.5B | 63.7 | 54.1 | 56.4 | 58.1 | – |
| Janus | 1.3B | 75.5 | 47.7 | 62.1 | 61.8 | – |
| Liquid | 7B | 71.5 | 52.3 | 65.1 | 63.0 | 30.7 |
| EMU3 | 8B | 61.1 | 47.3 | 61.9 | 56.8 | 31.3 |
| UniToken | 7B | 71.2 | 51.8 | 66.7 | 63.2 | – |
| **Uni-X$^\heartsuit$** | **3B / 4.5B** | **76.5** | **56.3** | **67.1** | **66.6** | **31.8** |

duplicated transformer is originally designed for image generation through diffusion. To remove the influence of diffusion and isolate architectural effects, we reconfigure the duplicated transformer to operate directly on image tokens. In this setup, the `qkv` sequences from the two transformers are concatenated within the attention module, allowing the model to incorporate visual information for both understanding and generation tasks. As a result, the MoT results reported in this paper reflect its effectiveness strictly within the VQ+AR paradigm, eliminating confounding factors introduced by diffusion-based processes.

### A.4 MORE EVALUATION RESULTS ON IMAGE GENERATION BENCHMARK

We conducted tests on the T2I-CompBench (Huang et al., 2025) and MSCOCO (Lin et al., 2015). Part of the results were excerpted from UniToken (Jiao et al., 2025). As shown in Table 7, Uni-X surpassed the recent strong autoregressive models EMU3 and Liquid in the newly added image generation benchmark. Uni-X also achieved better results than UniToken, which includes semantic information.

### A.5 GRADIENT CONFLICT ANALYSIS

**Analysis on Mainstream Models.** We further demonstrate the effectiveness of the current gradient conflict metric through experiments. We conduct a quantitative analysis on mainstream models

Table 8: Average gradient conflict between different domain data. Higher values indicate a higher degree of conflict.

| Model | Code vs. Math | Code vs. Wiki | Math vs. Wiki |
|---|---|---|---|
| Qwen2.5-1.5B | 0.158 | 0.330 | 0.262 |
| Qwen2.5-3B | 0.130 | 0.382 | 0.294 |
| Qwen2.5-Coder-3B | 0.182 | 0.317 | 0.275 |
| Qwen2.5-7B | 0.153 | 0.263 | 0.240 |
| Llama3.2-3B | 0.297 | 0.351 | 0.360 |

Table 9: Domain performance of Qwen2.5-3B and Qwen2.5-Coder-3B under zero-shot settings.

| Model | HumanEval (Code) | GSM8K (Math) | MMLU (Wiki) |
|---|---|---|---|
| Qwen2.5-3B | 39.0 | 6.0 | 65.0 |
| Qwen2.5-Coder-3B | 45.7 | 26.1 | 60.8 |

such as Qwen and Llama, as shown in Table 8. For each dataset, we utilized a total of 2M tokens (accumulated over 60 batches) to compute gradients, to ensure minimal gradient noise.

All models in Table 8 exhibit a consistent pattern: the gradient conflict between Code vs. Math is strictly lower than for both Code vs. Wiki and Math vs. Wiki. It is well-established in LLM pre-training that Code and Math tasks often mutually enhance each other (Shao et al., 2024; Ma et al., 2023). This phenomenon is precisely reflected in our gradient conflict analysis.

The relatively high gradient similarity (low conflict) between these two tasks implies that improvements in Code performance can drive improvements in Math performance. We further verified this in Table 9. Qwen2.5-Coder-3B, which was fine-tuned from Qwen2.5-3B to specifically enhance coding capabilities, simultaneously achieved a substantial improvement in Math performance. This validates our hypothesis that lower gradient conflict correlates with positive transfer between modalities/domains.

**Analysis on Other Modules.** In Section 3.1 of the main text, we analyzed gradient conflicts in the down-projection weights of the Feed-Forward Network (FFN). To develop a more complete picture and confirm that this issue is not confined to a single component, we extend our analysis to additional modules of the transformer. In particular, we examine gradient conflicts in the output projection weights (O_PROJ) and value projection weights (V_PROJ) of the self-attention mechanism, both of which play critical roles in multimodal representation learning.

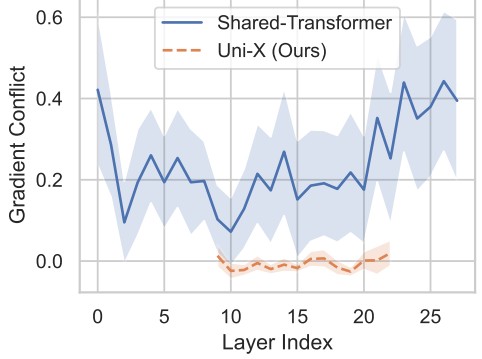 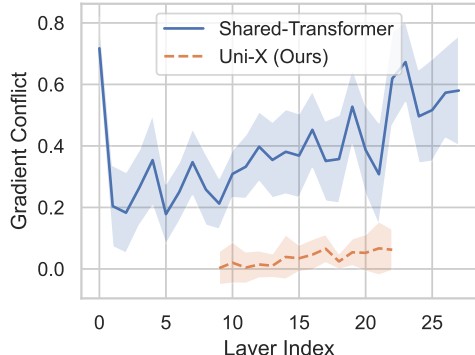

Figure 7: An analysis of gradient conflict in attention of out projection weights.

Figure 8: An analysis of gradient conflict in attention of value projection weights.

Using the same methodology for conflict measurement, Figures 7 and 8 reveal a consistent trend with that observed in the FFN layers. The modality-shared transformer exhibits severe gradient conflicts in the shallow and deep layers of both the attention output and value projection weights, with only partial alleviation in the middle layers. In contrast, Uni-X effectively addresses these issues: (i)

Table 10: Inference throughput comparison. Settings: batch size 48, input length ≈1,200 tokens, outputting one image.

| Model | # Params. | Tokens/s | Images/min |
|---|---|---|---|
| Shared Transformer (Qwen2.5-3B) | 3B | 975.2 | - |
| Liquid | 7B | 182.0 | 10.6 |
| EMU3 | 8B | 199.0 | 2.9 |
| **Uni-X** | 3B / 4.5B | 910.2 | **53.3** |

Table 11: Ratio between t-layers and v-layers within the separated layers.

| Configuration | MMLU | GenEval | MMB | Avg. ↑ |
|---|---|---|---|---|
| 14:8 | 48.2 | 37.8 | 26.1 | 37.4 |
| 14:14 | 49.6 | 41.3 | 29.4 | 40.1 |
| 14:20 | 50.1 | 42.6 | 31.0 | 41.2 |

modality-specific layers at both ends prevent conflicts in low-level processing and output stages, and (ii) the shared middle block further reduces residual conflicts by leveraging semantic alignment.

These results strengthen our hypothesis that gradient conflict stems from the intrinsic statistical mismatch between vision and text, and they demonstrate that Uni-X's two-end-separated, middle-shared design offers a robust and generalizable solution across multiple transformer components.

## A.6 INFERENCE EFFICIENCY

We have conducted a comprehensive evaluation of inference efficiency on an H800 PCIe (350W) GPU. As shown in Table 10, Uni-X demonstrates superior throughput compared to standard autoregressive baselines. Uni-X achieves high throughput (910.2 tokens/s) even compared to the original Qwen2.5-3B (975.2 tokens/s), despite the architectural changes and a higher number of parameters (4.5B vs 3B). This efficiency gain stems from the computational complexity of the attention mechanism in the separated layers.

Theoretically, the computational cost of the Uni-X architecture is lower, and the current inference speed still has a slight gap because the current code has not been fully optimized. In the separated layers, a sequence of length $n$ is effectively partitioned into vision tokens of length $a$ and text tokens of length $b$ (where $a + b = n$). Since the self-attention complexity is $O(n^2)$, and the separated layers enforce strict modality isolation, the complexity reduces to proportional to $a^2 + b^2$. Since $a^2 + b^2 < (a+b)^2 = n^2$, the computational cost for attention in these specific layers is strictly lower than in a fully shared transformer, leading to the observed speedup.

## A.7 ABLATION STUDY ON RATIO BETWEEN TEXT AND VISION.

We conducted experiments maintaining the same hyperparameters and training volume as in Table 5, and the results are shown in Table 11. We continued to use Qwen2.5-1.5B with a total of 28 layers as the base model. The number of vision layers directly affects the performance related to image understanding and generation. Surprisingly, reducing the number of vision layers also decreases pure text performance. This may be because the shared layers in the middle have to process more low-level vision information, thereby leading to a decline in pure text capability. This experimental result also proves the effectiveness of our proposed architecture from another perspective.

## A.8 USE OF LARGE LANGUAGE MODELS

Large Language Models (LLMs) were used solely as writing aids during manuscript preparation. Their role was limited to language polishing, improving grammar, clarity, and readability, without influencing the conceptual design, experimental methodology, or analytical findings. All research ideas, model designs, and experimental results are the original contributions of the authors.

