# OpenReview forum: "Uni-X: Mitigating Modality Conflict with a Two-End-Separated Architecture for Unified Multimodal Models"
_ICLR.cc/2026/Conference — ICLR 2026 Poster_

### Official Review · Reviewer_xf1h · 2025-10-28

**Soundness:** 3
**Presentation:** 2
**Contribution:** 4
**Rating:** 8
**Confidence:** 4

**Summary:**

This paper proposes Uni-X, a two-ended, middle-shared architecture: modality-specific blocks at the input and output are paired with a shared intermediate trunk for middle-level semantic fusion. This X-shaped layout mitigates gradient conflicts at both ends and further reduces residual interference within the shared layers. The design offers a clear architectural principle for unified generation models and merits wider exploration. Extensive experiments validate its effectiveness. Under identical training settings, Uni-X trains more efficiently than strong baselines. Moreover, the architecture scales to larger model sizes while delivering performance competitive with established large models.

**Strengths:**

Uni-X’s strengths lie in a clear diagnosis of why shared autoregressive multimodal transformers struggle and a simple remedy that matches the problem: the paper pinpoints where vision–text conflicts arise across depth, then proposes a two-end-separated, middle-shared
``X-shape'' layout that keeps early and late processing modality-specific while fusing semantics in the middle, yielding a lightweight design that reduces conflict without adding heavy subsystems. Empirically, the model shows strong results across understanding and generation, improves training throughput under matched settings, and scales efficiently, with a 3B variant achieving performance competitive with or better than several 7B baselines, which supports the claim that the approach is both practical and parameter-efficient for unified modeling.

**Weaknesses:**

The paper's weaknesses center on evidentiary clarity and completeness: the gradient-conflict metric $c_g$ is ambiguously defined (the roles of $g_{\text{any}}^{1}$ and $g_{\text{any}}^{2}$, indexing, aggregation, and normalization are unclear); inference efficiency is not evaluated, even though the X-shaped split-and-share design may impede parallelism relative to fully shared Transformers; Table 4 omits the exact number of shared layers and split points needed for replication; and the in-context learning claim is supported only by qualitative figures without quantitative benchmarks. However, I do not view these as major issues. The paper offers a systematic analysis of gradient conflict in unified generation models and provides useful insights for the community, so I believe it is a valuable contribution that deserves encouragement.

**Questions:**

1. I am not convinced that the conflict factor $c_g$ is defined in a way that faithfully reflects gradient conflicts between image and text inputs. In particular, the meanings of $g_{\text{any}}^{1}$ and $g_{\text{any}}^{2}$ are unclear. Please provide precise definitions on what ``any'' indexes. Since $c_g$ underpins the paper’s claims, a clear mathematical specification would be very helpful.

2. I noticed that Table 3 reports training efficiency. What about inference efficiency? Compared with a standard fully shared Transformer, the X-shaped split-and-share design may be less friendly to parallelization. I recommend evaluating Uni-X at inference time and discussing any trade-offs in latency and accuracy.

3. What is the number of shard layers in Table 4?

4. Uni-X’s in-context learning capability is demonstrated only qualitatively in Figure 5. Could you provide quantitative or statistical results to substantiate this claim?

---

> ### Author Response · Authors · 2025-11-23
> **Response (1/2) to Reviewer xf1h**
>
> We sincerely appreciate your recognition of our work as a valuable contribution and your constructive feedback regarding the definitions, efficiency, and experimental details. We have added the new experiments and analysis to the paper. We have addressed your questions point-by-point below.
>
> > **W1, Q1: The definition of gradient conflict is unclear and does not provide sufficient insights and evidence.**
>
> Thank you for this insightful question. To clarify the gradient conflict, we added a illustration and rewrote Section 3.1. In Multi-Task Learning, using Cosine similarity to measure gradient conflict is a standard practice [1, 2].
>
> We are the first to transfer this to the UMMs domain, and by introducing a baseline conflict value $S_\texttt{base}$, we directly analyze the gradient conflict in each layer. Thereby we discovered the phenomenon where gradient conflict is severe at both ends and mitigated in the middle.
>
>
> Cosine similarity is selected as the core metric because it **directly quantifies the geometric alignment of optimization directions** between modalities. Below, we provide a detailed explanation from both optimization and comparability perspectives.
>
> **Geometric and Optimization Perspective:** In the Shared Transformer, the final update direction is the aggregate of gradients from text ($g_\texttt{text}$) and image ($g_\texttt{img}$). The cosine similarity measures the angle $\theta$ between these two gradient vectors:
>
> * **Positive Similarity ($0^\circ \le \theta < 90^\circ$):** The gradients point in consistent directions. Updating parameters along this direction reduces the loss for both modalities simultaneously (Constructive Interference).
> * **Negative Similarity ($90^\circ < \theta \le 180^\circ$):** The gradients point in opposing directions. This indicates **Gradient Conflict**. In the extreme case ($\theta \approx 180^\circ$), reducing the loss for one modality explicitly increases the loss for the other. Even in the general obtuse case ($< 0$), the component of one gradient projected onto the other is negative, causing "destructive interference" that slows down convergence and destabilizes the training of shared layers.
>
> **We further demonstrate the effectiveness of the current gradient conflict metric through experiments.** We conduct a quantitative analysis on mainstream models such as Qwen and Llama, as shown in Table R1. For each dataset, we utilized a total of 2M tokens (accumulated over 60 batches) to compute gradients, to ensure minimal gradient noise.
>
> All models in Table R1 exhibit a consistent pattern: the gradient conflict between **Code vs. Math** is strictly **lower than** for both **Code vs. Wiki** and **Math vs. Wiki**. It is well-established in LLM pre-training that Code and Math tasks often mutually enhance each other [3, 4]. This phenomenon is precisely reflected in our gradient conflict analysis.
>
> The relatively high gradient similarity (low conflict) between these two tasks implies that improvements in Code performance can drive improvements in Math performance. We further verified this in Table R2. Qwen2.5-Coder-3B, which was fine-tuned from Qwen2.5-3B to specifically enhance coding capabilities, **simultaneously achieved a substantial improvement in Math performance**. This validates our hypothesis that lower gradient conflict correlates with positive transfer between modalities/domains. We have added this part of the experiment to Appendix A.4.
>
> ***Table R1**: Average Gradient Conflict between Different Domain Data. Higher values indicate a higher degree of conflict.*
>
> | Model            | Code vs. Math | Code vs. Wiki | Math vs. Wiki |
> | ---------------- | ------------- | ------------- | ------------- |
> | Qwen2.5-1.5B     | 0.158         | 0.330         | 0.262         |
> | Qwen2.5-3B       | 0.130         | 0.382         | 0.294         |
> | Qwen2.5-Coder-3B | 0.182         | 0.317         | 0.275         |
> | Qwen2.5-7B       | 0.153         | 0.263         | 0.240         |
> | Llama3.2-3B      | 0.297         | 0.351         | 0.360         |
>
> ***Table R2**: Domain Performance of Qwen2.5-3B and Qwen2.5-Coder-3B under zero-shot settings.*
>
> | Model            | HumanEval (Code) | GSM8K (Math) | MMLU (Wiki) |
> | ---------------- | ---------------- | ------------ | ----------- |
> | Qwen2.5-3B       | 39.0             | 6.0          | 65.0        |
> | Qwen2.5-Coder-3B | 45.7             | 26.1         | 60.8        |

---

> ### Author Response · Authors · 2025-11-23
> **Response (2/2) to Reviewer xf1h**
>
> >  **W2, Q2: Inference efficiency evaluation.**
>
> This is a very constructive suggestion. We have conducted a comprehensive evaluation of inference efficiency on an **H800 PCIe (350W)** GPU. As shown in Table R3, Uni-X demonstrates superior throughput compared to standard autoregressive baselines. Uni-X exhibits good inference efficiency, being $18\times$ faster than EMU3's image generation. We have added this part of the experiment to Appendix A.5.
>
> ***Table R3**: Inference Throughput Comparison. Settings: Batch size 48, input length $\approx$ 1200 tokens, outputting one image.*
>
> | Model | # Params. | Throughput (Tokens / Second) $\uparrow$ | Throughput (Images / Minute) $\uparrow$ |
> | :--- | ---- | :--- | :--- |
> | Fully Shared Transformer (Origin Qwen2.5-3B) | 3B | 975.2 | - |
> | **Uni-X** | 3B / 4.5B | 910.2 | **53.3** |
> | Liquid | 7B | 182.0 | 10.6 |
> | EMU3 | 8B | 199.0 | 2.9 |
>
> Uni-X achieves **high throughput (910.2 tokens/s)** even compared to the original Qwen2.5-3B (975.2 tokens/s), despite the architectural changes and a higher number of parameters (4.5B vs 3B). This efficiency gain stems from the computational complexity of the attention mechanism in the separated layers.
>
> **Theoretically, the computational cost of the Uni-X architecture is lower**, and the current inference speed still has a slight gap because the current code has not been fully optimized. In the separated layers, a sequence of length $n$ is effectively partitioned into vision tokens of length $a$ and text tokens of length $b$ (where $a+b=n$). Since the self-attention complexity is $ O(n^2) $, and the separated layers enforce strict modality isolation, the complexity reduces to proportional to $a^2 + b^2$. Since $ a^2 + b^2 < (a+b)^2=n^2 $, the computational cost for attention in these specific layers is strictly lower than in a fully shared transformer, leading to the observed speedup.
>
>
>
>
> >  **W3, Q3: Number of shared layers in Table 4.**
>
> Thank you for pointing this out, we have updated this in the paper. Our ablation studies are based on the **Qwen2.5-1.5B** architecture, which consists of **28 layers**. The number of shared layers is calculated as:
> $N_\texttt{shared} = N_\texttt{total} - N_\texttt{shallow-split} - N_\texttt{deep-split}$
>
> For a configuration denoted as **$ x:y $** (where $x$ is the number of shallow separated layers and $y$ is the number of deep separated layers), the layer distribution is:
> * Shallow Separated Layers: Layers $[0, x)$
> * Shared Layers: Layers $[x, 28 - y)$
> * Deep Separated Layers: Layers $[28 - y, 28)$
>
> Taking the optimal **9:5** configuration as an example:
> * Shallow Separated: 9 layers.
>
> * Deep Separated: 5 layers.
>
> * Shared Layers: $28 - 9 - 5 = \mathbf{14}$ layers.
>
>
> > **W4, Q4: Quantitative results for in-context learning (ICL).**
>
> Thank you for your suggestion. Quantifying in-context learning capability is indeed a very valuable experiment. The primary focus of this paper is to establish Uni-X as a robust foundation for mitigating gradient conflicts in standard training and generation tasks. The ICL capability shown is an emergent property, which is particularly notable given that Uni-X was trained on non-interleaved data. While we currently focus on the architectural validation of the "X-shape" design, we agree that rigorous quantitative benchmarking of ICL is an important next step for this line of research.
>
> [1] Gradient Surgery for Multi-Task Learning
>
> [2] Recon: Reducing Conflicting Gradients from the Root for Multi-Task Learning
>
> [3] At Which Training Stage Does Code Data Help LLMs Reasoning?
>
> [4] DeepSeekMath: Pushing the Limits of Mathematical Reasoning in Open Language Models

---

> > ### Comment · Reviewer_xf1h · 2025-11-24
> >
> > Thank you for the detailed rebuttal and the clarifications provided. After carefully considering your responses and additional evidence, I am satisfied with how the main concerns have been addressed and have decided to keep my positive overall rating for this submission.

---

> ### Author Response · Authors · 2025-11-24
>
> Thank you again for your thorough and constructive review. We are delighted that our response addressed your concerns.

---

### Official Review · Reviewer_7qva · 2025-10-29

**Soundness:** 3
**Presentation:** 3
**Contribution:** 3
**Rating:** 4
**Confidence:** 3

**Summary:**

This paper propose Uni-X, a two-end-separated, middle-shared architecture, whose X-shaped design not only eliminates gradient conflicts at both ends but also further alleviates residual conflicts in the shared layers. When scaled to 3B parameters with larger training data, Uni-X matches or surpasses 7B AR-based UMMs, achieving a GenEval score of 82 for image generation alongside strong performance in text and vision understanding tasks. These results establish Uni-X as a parameter-efficient and scalable foundation for future unified multimodal modeling.

**Strengths:**

1. It proposes the Uni-X architecture, which processes vision and text inputs separately, fuses them for computation only in the intermediate layers, and then separates them again for modality-specific outputs.
2. It defines the concept of "Gradient Conflict" to represent the conflict between the vision and text components during model training. This concept is highly extensible to other multimodal models for analyzing the interplay between different modalities, making it valuable for future research in the field.

**Weaknesses:**

1. The paper defines *Gradient Conflict*, but the later sections lack further analysis. What I would like to understand is how Gradient Conflict reduces training efficiency or how it leads to model failures during inference. Is Gradient Conflict severe in mainstream models such as Qwen? Specifically, I hope to see a quantitative comparison of Gradient Conflict across different models and an analysis of its impacts.

2. The experimental results in the paper show that Uni-X is comparable to other large models, but we notice that Uni-X performs worse than its base model Qwen2.5-3B on multiple benchmarks such as ARC-C and MMLU. I am concerned about whether such a Uni-X architecture is reasonable, and I hope to see further analysis of this phenomenon.

**Questions:**

1. The paper defines  $S_{base} = \cos(g^1_{any}, g^2_{any})$ . How exactly are ( $g^1_{any}$ ) and ( $g^2_{any}$ ) computed? Are they averaged over multiple batches? For different types of multimodal data such as understanding, captioning, and generation, is ( $S_{base}$ ) stable? In other words, is this an inherent internal characteristic of the model?

2. The paper compares the influence of different ratios between shallow separated layers ($x$) and deep separated layers ($y$). Another point of interest is the parameter ratio between t-layers and v-layers within the separated layers. I would like to see an ablation study regarding this aspect.

In summary, my primary concerns lie in **Weaknesses 1 and Weaknesses 2**. If the responses are reasonable and acceptable, I will raise the score.

---

> ### Author Response · Authors · 2025-11-23
> **Response (1/2) to Reviewer 7qva**
>
> We sincerely thank you for your insightful suggestions, which have significantly enhanced the quality of our paper. We have added the new experiments and analysis to the paper. Below, we address your concerns in detail.
>
> > **W1: Lack further analysis of Gradient Conflict. Is Gradient Conflict severe in mainstream models such as Qwen? Specifically, I hope to see a quantitative comparison of Gradient Conflict across different models and an analysis of its impacts.**
>
> We conduct a quantitative analysis of gradient conflict on mainstream models such as Qwen and Llama, as shown in Table R1. For each dataset, we utilized a total of 2M tokens (accumulated over 60 batches) to compute gradients, ensuring minimized gradient noise.
>
> All models in Table R1 exhibit a consistent pattern: the gradient conflict between **Code vs. Math** is strictly **lower than** that of **Code vs. Wiki** and **Math vs. Wiki**. It is well-established in LLM pre-training that Code and Math tasks often mutually enhance each other [1, 2]. This phenomenon is precisely reflected in our gradient conflict analysis.
>
> The relatively high gradient similarity (low conflict) between these two tasks implies that improvements in Code performance can drive improvements in Math performance. We further verified this in Table R2. Qwen2.5-Coder-3B, which was fine-tuned from Qwen2.5-3B to specifically enhance coding capabilities, **simultaneously achieved a substantial improvement in Math performance**. This validates our hypothesis that lower gradient conflict correlates with positive transfer between modalities/domains. We have added this part of the experiment to Appendix A.4.
>
> ***Table R1**: Average Gradient Conflict between Different Domain Data. Higher values indicate a higher degree of conflict.*
>
> | Model            | Code vs. Math | Code vs. Wiki | Math vs. Wiki |
> | ---------------- | ------------- | ------------- | ------------- |
> | Qwen2.5-1.5B     | 0.158         | 0.330         | 0.262         |
> | Qwen2.5-3B       | 0.130         | 0.382         | 0.294         |
> | Qwen2.5-Coder-3B | 0.182         | 0.317         | 0.275         |
> | Qwen2.5-7B       | 0.153         | 0.263         | 0.240         |
> | Llama3.2-3B      | 0.297         | 0.351         | 0.360         |
>
> ***Table R2**: Domain Performance of Qwen2.5-3B and Qwen2.5-Coder-3B under zero-shot settings.*
>
> | Model            | HumanEval (Code) | GSM8K (Math) | MMLU (Wiki) |
> | ---------------- | ---------------- | ------------ | ----------- |
> | Qwen2.5-3B       | 39.0             | 6.0          | 65.0        |
> | Qwen2.5-Coder-3B | 45.7             | 26.1         | 60.8        |
>
>
>
> > **Q1.1: How exactly are $g_{any}^1$ and $g_{any}^2$ computed? Are they averaged over multiple batches?**
>
> Text-only data ($D_{\texttt{text}}$) and image-text pair data ($D_{\texttt{img}}$) are both randomly sampled from the pre-training data. Arbitrary modality data ($D_{\texttt{any}}$) is also randomly sampled from the pre-training data **without considering the modality**. We randomly split $D_{\texttt{any}}$ evenly into two parts, $D_\texttt{any}^1$ and $D_\texttt{any}^2$, and use these two groups of data to calculate gradients separately, obtaining $g_\texttt{any}^1$ and $g_\texttt{any}^2$.
>
> In our analysis of gradient conflicts, all gradient data are averages over more than 60 mini-batches, and each subset ($D_{\texttt{text}}$, $D_{\texttt{img}}$ and $D_{\texttt{any}}$) contains more than 2M tokens.

---

> ### Author Response · Authors · 2025-11-23
> **Response (2/2) to Reviewer 7qva**
>
> > **Q1.2: For different types of multimodal data such as understanding, captioning, and generation, is ($S_{base}$) stable? In other words, is this an inherent internal characteristic of the model?**
>
> Because the $D_{\texttt{any}}$ data includes text-only, image understanding, image captioning, and image generation, the $S_{base}$ is stable. In other words, as you mentioned, it represents an inherent property of the model at that layer.
>
>
> > **W2: Performs worse than its base model Qwen2.5-3B on multiple text-only benchmarks.**
>
> We hypothesize that this performance gap is primarily **due to the quality of our pre-training text data being inferior to Qwen's proprietary annealing data**.
>
> We verified this through a controlled experiment: we performed continual training on the original Qwen2.5-3B model using **only the text component** of the Uni-X pre-training dataset (completely removing image modality data). Due to computational cost constraints, we trained for 12B tokens.
>
> As shown in Table R3, the results indicate that the model's performance on text benchmarks degrades as training progresses on our dataset. This suggests that the decline in Uni-X's text performance compared to the base model is likely attributable to the difference in text data quality rather than a flaw in the Uni-X architecture itself.
>
> ***Table R3**: Continual Training on Qwen2.5-3B using Uni-X Text Data.*
>
> | # Tokens          | ARC-E | ARC-C | WinoG | BoolQ | MMLU | Avg.$\uparrow$ |
> | ----------------- | ----- | ----- | ----- | ----- | ---- | -------------- |
> | Origin Qwen2.5-3B | 77.4  | 47.2  | 68.1  | 77.5  | 65.1 | 67.1           |
> | 6B                | 76.3  | 45.8  | 67.7  | 74.4  | 60.1 | 64.9           |
> | 12B               | 76.2  | 44.2  | 66.4  | 75.0  | 60.3 | 64.4           |
>
>
>
> > **Q2: Ratio between t-layers and v-layers within the separated layers.**
>
> Thank you very much for your insightful suggestions. The parameter ratio between t-layers and v-layers is indeed a highly valuable hyperparameter. We conducted experiments maintaining the same hyperparameters and training volume as in Table 5 of the paper, and the results are shown in Table R4. We continued to use Qwen2.5-1.5B with a total of 28 layers as the base model.
>
> The number of vision layers directly affects the performance related to image understanding and generation. Surprisingly, reducing the number of vision layers also decreases pure text performance. This may be because the shared layers in the middle have to process more low-level vision information, thereby leading to a decline in pure text capability. This experimental result also proves the effectiveness of our proposed architecture from another perspective. We have added this part of the experiment to Appendix A.6.
>
> ***Table R4**: Ratio between t-layers and v-layers within the separated layers.*
>
> | Text/Vision Layer Ratio | MMLU | GenEval | MMB  | Avg. $\uparrow$ |
> | ----------------------- | ---- | ------- | ---- | --------------- |
> | 14:8                    | 48.2 | 37.8    | 26.1 | 37.4            |
> | 14:14                   | 49.6 | 41.3    | 29.4 | 40.1            |
> | 14:20                   | 50.1 | 42.6    | 31.0 | 41.2            |
>
>
>
> [1] At Which Training Stage Does Code Data Help LLMs Reasoning?
>
> [2] DeepSeekMath: Pushing the Limits of Mathematical Reasoning in Open Language Models

---

> ### Author Response · Authors · 2025-11-28
>
> Dear reviewer 7qva,
>
> Thank you for your time reviewing, as the discussion closing date is December 2nd. We are very happy to provide more clarification if you have any unresolved or further questions or concerns.
>
> Sincerely,
>
> Authors

---

### Official Review · Reviewer_v2vp · 2025-10-30

**Soundness:** 2
**Presentation:** 3
**Contribution:** 2
**Rating:** 4
**Confidence:** 3

**Summary:**

The paper introduces a novel unified multimodal model architecture named Uni-X, which aims to address the gradient conflict issue in current unified multimodal models (UMMs) based on shared autoregressive (AR) Transformers. Uni-X employs an "X-shaped" structure design with "separation at both ends and sharing in the middle," effectively alleviating conflicts between the visual and textual modalities while maintaining model simplicity. This design enhances training efficiency and performance in generation and understanding.

**Strengths:**

The paper measures the degree of cross-modal gradient conflict by comparing the similarity of backpropagated gradients. It is found that visual and textual modalities cause severe gradient conflicts in both shallow and deep layers due to their vastly different underlying statistical properties, making it difficult for the model to optimize both simultaneously. In contrast, middle layers exhibit less conflict as they represent more abstract and semantically aligned features. Therefore, the authors propose designing model architectures based on modality characteristics rather than enforcing shared parameters across all layers. A series of experiments demonstrate the effectiveness of the proposed architecture.

The paper presents innovative insights, with solid experimental quality and clear expression, and holds significant importance.

**Weaknesses:**

1. The authors' explanation of using gradient similarity as a measure of gradient conflict is not particularly clear. Additional explanation is needed to clarify why lower similarity indicates stronger gradient conflict.

2. Uni-X modifies the fully shared network structure by introducing separate branches for vision and text. This seems to revert to the classical dual-tower structure, with separated shallow and deep layers seemingly taking on some of the functions of a multimodal encoder-decoder. The authors do not elaborate on the similarities and differences between the Uni-X structure and classical structure.

3. The paper lacks experimental comparisons with some recent models, such as blipo3-Next.

**Questions:**

1. Please explain why lower gradient similarity indicates stronger gradient conflict.

2. Please elaborate on the similarities and differences between the Uni-X structure and the dual-tower architecture. If gradient conflicts imply the separation of the model, does it mean that a dual-tower architecture rather than a fully shared architecture is more suitable for generating and understanding integrated tasks?

---

> ### Author Response · Authors · 2025-11-23
> **Response (1/2) to Reviewer v2vp**
>
> Thank you for your thoughtful and insightful feedback. We appreciate your recognition of our work as providing "innovative insights, with solid experimental quality and clear expression, and holding significant importance." We have added the new experiments to the paper. Please find our detailed responses below.
>
> > **W1, Q1: Explanation of using gradient similarity as a measure of gradient conflict is not particularly clear.**
>
> Thank you for this insightful question. To clarify the gradient conflict, we added a illustration and rewrote Section 3.1. In Multi-Task Learning, using Cosine similarity to measure gradient conflict is a standard practice [1, 2].
>
> We are the first to transfer this to the UMMs domain, and by introducing a baseline conflict value $S_\texttt{base}$, we directly analyze the gradient conflict in each layer. Thereby we discovered the phenomenon where gradient conflict is severe at both ends and mitigated in the middle.
>
> Cosine similarity is selected as the core metric because it **directly quantifies the geometric alignment of optimization directions** between modalities. Below, we provide a detailed explanation from both optimization and comparability perspectives.
>
> **Geometric and Optimization Perspective:** In the Shared Transformer, the final update direction is the aggregate of gradients from text ($g_\texttt{text}$) and image ($g_\texttt{img}$). The cosine similarity measures the angle $\theta$ between these two gradient vectors:
>
> * **Positive Similarity ($0^\circ \le \theta < 90^\circ$):** The gradients point in consistent directions. Updating parameters along this direction reduces the loss for both modalities simultaneously (Constructive Interference).
> * **Negative Similarity ($90^\circ < \theta \le 180^\circ$):** The gradients point in opposing directions. This indicates **Gradient Conflict**. In the extreme case ($\theta \approx 180^\circ$), reducing the loss for one modality explicitly increases the loss for the other. Even in the general obtuse case ($< 0$), the component of one gradient projected onto the other is negative, causing "destructive interference" that slows down convergence and destabilizes the training of shared layers.
>
> **We further demonstrate the effectiveness of the current gradient conflict metric through experiments.** We conduct a quantitative analysis on mainstream models such as Qwen and Llama, as shown in Table R1. For each dataset, we utilized a total of 2M tokens (accumulated over 60 batches) to compute gradients, to ensure minimal gradient noise.
>
> All models in Table R1 exhibit a consistent pattern: the gradient conflict between **Code vs. Math** is strictly **lower than** for both **Code vs. Wiki** and **Math vs. Wiki**. It is well-established in LLM pre-training that Code and Math tasks often mutually enhance each other [3, 4]. This phenomenon is precisely reflected in our gradient conflict analysis.
>
> The relatively high gradient similarity (low conflict) between these two tasks implies that improvements in Code performance can drive improvements in Math performance. We further verified this in Table R2. Qwen2.5-Coder-3B, which was fine-tuned from Qwen2.5-3B to specifically enhance coding capabilities, **simultaneously achieved a substantial improvement in Math performance**. This validates our hypothesis that lower gradient conflict correlates with positive transfer between modalities/domains. We have added this part of the experiment to Appendix A.4.
>
> ***Table R1**: Average Gradient Conflict between Different Domain Data. Higher values indicate a higher degree of conflict.*
>
> | Model            | Code vs. Math | Code vs. Wiki | Math vs. Wiki |
> | ---------------- | ------------- | ------------- | ------------- |
> | Qwen2.5-1.5B     | 0.158         | 0.330         | 0.262         |
> | Qwen2.5-3B       | 0.130         | 0.382         | 0.294         |
> | Qwen2.5-Coder-3B | 0.182         | 0.317         | 0.275         |
> | Qwen2.5-7B       | 0.153         | 0.263         | 0.240         |
> | Llama3.2-3B      | 0.297         | 0.351         | 0.360         |
>
> ***Table R2**: Domain Performance of Qwen2.5-3B and Qwen2.5-Coder-3B under zero-shot settings.*
>
> | Model            | HumanEval (Code) | GSM8K (Math) | MMLU (Wiki) |
> | ---------------- | ---------------- | ------------ | ----------- |
> | Qwen2.5-3B       | 39.0             | 6.0          | 65.0        |
> | Qwen2.5-Coder-3B | 45.7             | 26.1         | 60.8        |

---

> ### Author Response · Authors · 2025-11-23
> **Response (2/2) to Reviewer v2vp**
>
> > **W2: Similarities and differences between the Uni-X structure and classical dual-tower structure. Separated shallow and deep layers are taking on some of the functions of a multimodal encoder-decoder.**
>
> We thank the reviewer for this insightful observation. While Uni-X shares the concept of **modality separation** with dual-tower architectures, it differs fundamentally in three key aspects:
>
> 1.  **Different Objectives :** Dual-tower models focus on *Representation Alignment* for retrieval/classification. Uni-X is a *Unified Generative Model*.
> 1.  **Middle-Sharing for Semantic Fusion:** unlike classical dual-tower models (e.g., CLIP) that maintain separation until the final metric alignment, Uni-X shares the intermediate layers. Our core insight is that while low-level statistical properties differ, **high-level semantics are naturally aligned**.
>     * ***Evidence**:* Our ablation study (Table 5 of main paper) demonstrates that reducing the number of shared layers (moving closer to a full dual-tower) actually degrades performance, proving the necessity of the shared semantic core.
> 3.  **Single AR Framework:** Uni-X is not an encoder-decoder architecture (which typically connects a bidirectional encoder to an autoregressive decoder via cross-attention). Instead, it remains a pure decoder-only autoregressive framework. This structural simplicity makes it easy to adapt to existing inference frameworks like vLLM and SGLang, and also makes it easy to further extend its interleaved image-text generation capabilities in the future.
>
> **Regarding your observation on "encoder/decoder" roles:** We fully agree. The separated shallow and deep layers effectively function as lightweight, modality-specific "encoders" and "decoders." However, a key advantage of Uni-X is that these modules are **learned end-to-end** within the model, ensuring better coherence between perception and generation.
>
>
>
>
> > **W3: Lacks experimental comparisons with some recent models.**
>
> Thank you for sharing this *latest* work with us. As shown in Table R3, we compared it with Blip3o-Next, and its performance was slightly inferior. However, we believe this result is acceptable.
>
> - Blip3o-Next used GenEval as a reward for reinforcement learning, which might lead to an inflated score on GenEval.
> - Blip3o-Next used 10 million samples for training on graphic editing tasks, while Uni-X only used 90k, but still achieved comparable performance.
> - Blip3o-Next uses a pre-trained diffusion backbone SANA-1.5, which to some extent is equivalent to performing additional training on image generation tasks.
>
>
>
> ***Table R3**: Comparison between Uni-X and Blip3o-Next.*
>
> | Model       | GenEval | ImgEdit |
> | ----------- | ------- | ------- |
> | Blip3o-Next | 91      | 3.62    |
> | Uni-X       | 83      | 3.44    |
>
>
>
>
> > **Q2: Is dual-tower architecture more suitable than a fully shared architecture for generating and understanding integrated tasks.**
>
> This is a very constructive question. As shown in Table 5 of the main paper, although reducing the number of intermediate layers results in **more parameters, the performance actually decreases**. This indicates that neither a fully shared architecture nor a dual-tower architecture is well suited for unified understanding and generation tasks. Uni-X provides a trade-off between these two approaches by separating the most conflicting regions (the two ends). As you mentioned above, these split layers actually serve as partial encoder and decoder functions. At the same time, it retains the “middle shared” layers, which are crucial for the model to learn cross-modal abstraction and high-level semantics.
>
>
>
> [1] Gradient Surgery for Multi-Task Learning
>
> [2] Recon: Reducing Conflicting Gradients from the Root for Multi-Task Learning
>
> [3] At Which Training Stage Does Code Data Help LLMs Reasoning?
>
> [4] DeepSeekMath: Pushing the Limits of Mathematical Reasoning in Open Language Models

---

> ### Author Response · Authors · 2025-11-28
>
> Dear reviewer v2vp,
>
> Thank you for your time reviewing, as the discussion closing date is December 2nd. We are very happy to provide more clarification if you have any unresolved or further questions or concerns.
>
> Sincerely,
>
> Authors

---

### Official Review · Reviewer_z7aM · 2025-10-31

**Soundness:** 3
**Presentation:** 2
**Contribution:** 2
**Rating:** 4
**Confidence:** 3

**Summary:**

This paper introduces Uni-X, a new unified mllm architecture that addresses the "gradient conflict" phenomenon during unified training. By separating first and final layers while fusing different modalities in the middle layers, conflited gradients are alleviated compared to standard architectures. By scaling on data and compute, Uni-X achieves good performance in generation and understanding compared to its counterparts.

**Strengths:**

1. The presentation is clear and the paper is easy to follow.
2. The code is already open-sourced, which is a plus for the community.

**Weaknesses:**

1. More evaluations on image generation benchmarks, e.g., MSCOCO, T2I-CompBench, should be considered to further validate the effectiveness of Uni-X.
2. Some baselines like Blip3o and Bagel are also evaluated on image editing benchmarks like ImgEdit or OmniContext, why is Uni-X not evaluated on such benchmarks? I believe if Uni-X is more modality unified as claimed, it should be competent in tasks like image editing that requires both image and text understanding.
3. The overall performance of Uni-X is mediocore compared with other baselines.

**Questions:**

1. Why does the authors choose cosine similarity as metrics for gradient-conflict? I believe more insights on this would be beneficial.

---

> ### Author Response · Authors · 2025-11-23
> **Response (1/3) to Reviewer z7aM**
>
> Thank you very much for your constructive feedback. We appreciate your recognition of our clear presentation and the contribution of our open-source code to the community. We have added the new experiments and analysis to the paper. Below, we address your concerns in detail.
>
> > **W1: More evaluations on image generation benchmarks, e.g., MSCOCO, T2I-CompBench.**
>
> We conducted tests on the two benchmarks you mentioned. Some of the results were excerpted from UniToken [1]. As shown in *Table R1*, **Uni-X surpassed the recent strong autoregressive models EMU3 and Liquid** in the newly added image generation benchmark. Uni-X also achieved better results than UniToken, which includes semantic information. We have added this part of the experiment to Appendix A.7.
>
> ***Table R1**: Results on T2I-CompBench and MSCOCO.*
>
> | Model | # Params. | T2I-Color | T2I-Shape | T2I-Texture | T2I-Avg. $\uparrow$ | MSCOCO CLIP-T $\uparrow$ |
> | :-- | :-- | --: | --: | --: | --: | --: |
> | SDXL | 3.5B | 63.7 | 54.1 | 56.4 | 58.1 | - |
> | Janus | 1.3B | 75.5 | 47.7 | 62.1 | 61.8 | - |
> | Liquid | 7B | 71.5 | 52.3 | 65.1 | 63.0 | 30.7 |
> | EMU3 | 8B | 61.1 | 47.3 | 61.9 | 56.8 | 31.3 |
> | UniToken | 7B | 71.2 | 51.8 | 66.7 | 63.2 | - |
> | **Uni-X** | 3B / 4.5B | **76.5** | **56.3** | **67.1** | **66.6** | **31.8** |
>
>
>
>
> > **W2: More evaluations on image editing benchmarks like ImgEdit or OmniContext.**
>
> Your advice is very insightful. We conducted tests on ImgEdit, and the results are shown in Table R2. **Uni-X achieved better results than Bagel**, even with less training data and fewer parameters than Bagel. This demonstrates that the high-level semantic unification of Uni-X greatly enhances its image editing capabilities. Specifically, we fine-tuned Uni-X on the ShareGPT4o and subset of GPT-Image-Edit image editing data, totaling only 90k datasets for 2 epochs. We have added this part of the experiment to Table 3 of the main paper.
>
> ***Table R2**: Results on ImgEdit.*
>
> | Model       | # Params. |  Add | Adjust | Extract | Replace | Remove | Background | Style | Hybird | Action | Overall$\uparrow$ |
> | :---------- | :-------- | ---: | -----: | ------: | ------: | -----: | ---------: | ----: | -----: | -----: | ----------------: |
> | GPT-4o      | -         | 4.61 |   4.33 |    2.90 |    4.35 |   3.66 |       4.57 |  4.93 |   3.96 |   4.89 |          **4.20** |
> | ICEdit      | 12B       | 3.58 |   3.39 |    1.73 |    3.15 |   2.93 |       3.08 |  3.84 |   2.04 |   3.68 |              3.05 |
> | AnyEdit     | 4B        | 3.18 |   2.95 |    1.88 |    2.47 |   2.23 |       2.24 |  2.85 |   1.56 |   2.65 |              2.45 |
> | UltraEdit   | 4B        | 3.44 |   2.81 |    2.13 |    2.96 |   1.45 |       2.83 |  3.76 |   1.91 |   2.98 |              2.70 |
> | Step1X-Edit | 12B       | 3.88 |   3.14 |    1.76 |    3.40 |   2.41 |       3.16 |  4.63 |   2.64 |   2.52 |              3.06 |
> | Bagel       | 7B / 14B  | 3.56 |   3.31 |    1.70 |    3.30 |   2.62 |       3.24 |  4.49 |   2.38 |   4.17 |              3.20 |
> | **Uni-X**   | 3B / 4.5B | 3.57 |   3.18 |    2.06 |    3.94 |   3.82 |       3.38 |  4.21 |   3.16 |   3.63 |       3.44 |

---

> ### Author Response · Authors · 2025-11-23
> **Response (2/3) to Reviewer z7aM**
>
> > **W3: Performance of Uni-X is mediocore compared with other baselines.**
>
> Table 2 in the paper shows that Uni-X **has already achieved competitive performance** despite using significantly fewer parameters and training data than the baselines. To further demonstrate scalability, we train Uni-X$^{\heartsuit} $ with additional image understanding and captioning data (totaling 240B tokens). This proves that understanding ability can be further improved by increasing the amount of data.
>
> As shown in Table R3, Uni-X demonstrates **SOTA image generation performance** among autoregressive unified multi-modal models (UMMs) without semantic encoders, despite using considerably **fewer parameters and training data**.
>
> Uni-X performs slightly worse in understanding tasks compared to EMU3, primarily because EMU3 encodes $512 \times 512$ images into a token sequence length of $4096$, whereas Uni-X's encoding length is only $1024$. The longer sequence in EMU3 allows for more detailed information in understanding tasks but **significantly reduces inference efficiency**, decreasing image generation efficiency by more than $18\times$.
>
>
>
> ***Table R3**: Comparison with AR UMMs w/o semantic encoders. Uni-X$^{\heartsuit}$ denotes the model trained with increased data (240B tokens).*
>
>
> | Model                    | # Tokens | # Params. | Throughput (Images/Min) $\uparrow$ | GenEval | DPG  | T2I-CompBench | MSCOCO | MME     | POPE | MMB  | SEED |
> | :----------------------- | :------- | :-------- | :-------------------------------------- | :------ | :--- | :------------ | :----- | :------ | :--- | :--- | :--- |
> | Chameleon                | 9.2T     | 34B       | -                                       | 39      | -    | -             | -      | 604.5   | -    | 32.7 | -    |
> | LWM                      | ~500B    | 7B        | -                                       | 47      | -    | -             | -      | -       | 75.2 | -    | -    |
> | EMU3                     | -        | 8B        | 2.9                                     | 66      | 80.6 | 56.8          | 31.3   | 1,243.8 | 85.2 | 58.5 | 68.2 |
> | Liquid                   | ~90B     | 7B        | 10.6                                    | 68      | 79.8 | 63.0          | 30.7   | 1,107.2 | 81.1 | -    | -    |
> | **Uni-X**                | 140B     | 3B / 4.5B | 53.3                                    | 82      | 79.8 | -             | -      | 1,158.3 | 83.6 | 59.3 | 60.2 |
> | **Uni-X$^{\heartsuit}$** | 240B     | 3B / 4.5B | 53.3                                    | 83      | 80.3 | 66.6          | 31.8   | 1,228.2 | 84.6 | 62.7 | 59.8 |
>
>
> Compared to other diffusion models like Bagel (5.1T tokens), Uni-X achieves superior results with <5% of the training data and ~32% of the parameters (3B/4.5B vs 7B/14B), proving it is highly efficient rather than mediocre. Models using pre-trained vision encoders (e.g., CLIP/SigLIP) benefit from external semantic shortcuts. In the pure AR category, where models must learn visual tokens from scratch, Uni-X performs well.

---

> ### Author Response · Authors · 2025-11-23
> **Response (3/3) to Reviewer z7aM**
>
> > **Q1: Why does the authors choose cosine similarity as metrics for gradient-conflict? I believe more insights on this would be beneficial.**
>
> Thank you for this insightful question. To clarify the gradient conflict, we added a illustration and rewrote Section 3.1. In Multi-Task Learning, using Cosine similarity to measure gradient conflict is a standard practice [1, 2].
>
> We are the first to transfer this to the UMMs domain, and by introducing a baseline conflict value $S_\texttt{base}$, we directly analyze the gradient conflict in each layer. Thereby we discovered the phenomenon where gradient conflict is severe at both ends and mitigated in the middle.
>
> Cosine similarity is selected as the core metric because it **directly quantifies the geometric alignment of optimization directions** between modalities. Below, we provide a detailed explanation from both optimization and comparability perspectives.
>
> **Geometric and Optimization Perspective:** In the Shared Transformer, the final update direction is the aggregate of gradients from text ($g_\texttt{text}$) and image ($g_\texttt{img}$). The cosine similarity measures the angle $\theta$ between these two gradient vectors:
>
> *   **Positive Similarity ($0^\circ \le \theta < 90^\circ$):** The gradients point in consistent directions. Updating parameters along this direction reduces the loss for both modalities simultaneously (Constructive Interference).
> *   **Negative Similarity ($90^\circ < \theta \le 180^\circ$):** The gradients point in opposing directions. This indicates **Gradient Conflict**. In the extreme case ($\theta \approx 180^\circ$), reducing the loss for one modality explicitly increases the loss for the other. Even in the general obtuse case ($< 0$), the component of one gradient projected onto the other is negative, causing "destructive interference" that slows down convergence and destabilizes the training of shared layers.
>
> **We further demonstrate the effectiveness of the current gradient conflict metric through experiments.** We conduct a quantitative analysis on mainstream models such as Qwen and Llama, as shown in Table R4. For each dataset, we utilized a total of 2M tokens (accumulated over 60 batches) to compute gradients, to ensure minimal gradient noise.
>
> All models in Table R4 exhibit a consistent pattern: the gradient conflict between **Code vs. Math** is strictly **lower than** for both **Code vs. Wiki** and **Math vs. Wiki**. It is well-established in LLM pre-training that Code and Math tasks often mutually enhance each other [4, 5]. This phenomenon is precisely reflected in our gradient conflict analysis.
>
> The relatively high gradient similarity (low conflict) between these two tasks implies that improvements in Code performance can drive improvements in Math performance. We further verified this in Table R5. Qwen2.5-Coder-3B, which was fine-tuned from Qwen2.5-3B to specifically enhance coding capabilities, **simultaneously achieved a substantial improvement in Math performance**. This validates our hypothesis that lower gradient conflict correlates with positive transfer between modalities/domains. We have added this part of the experiment to Appendix A.4.
>
> ***Table R4**: Average Gradient Conflict between Different Domain Data. Higher values indicate a higher degree of conflict.*
>
> | Model            | Code vs. Math | Code vs. Wiki | Math vs. Wiki |
> | ---------------- | ------------- | ------------- | ------------- |
> | Qwen2.5-1.5B     | 0.158         | 0.330         | 0.262         |
> | Qwen2.5-3B       | 0.130         | 0.382         | 0.294         |
> | Qwen2.5-Coder-3B | 0.182         | 0.317         | 0.275         |
> | Qwen2.5-7B       | 0.153         | 0.263         | 0.240         |
> | Llama3.2-3B      | 0.297         | 0.351         | 0.360         |
>
> ***Table R5**: Domain Performance of Qwen2.5-3B and Qwen2.5-Coder-3B under zero-shot settings.*
>
> | Model            | HumanEval (Code) | GSM8K (Math) | MMLU (Wiki) |
> | ---------------- | ---------------- | ------------ | ----------- |
> | Qwen2.5-3B       | 39.0             | 6.0          | 65.0        |
> | Qwen2.5-Coder-3B | 45.7             | 26.1         | 60.8        |
>
>
>
>
> [1] UniToken: Harmonizing Multimodal Understanding and Generation through Unified Visual Encoding
>
> [2] Gradient Surgery for Multi-Task Learning
>
> [3] Recon: Reducing Conflicting Gradients from the Root for Multi-Task Learning
>
> [4] At Which Training Stage Does Code Data Help LLMs Reasoning?
>
> [5] DeepSeekMath: Pushing the Limits of Mathematical Reasoning in Open Language Models

---

> > ### Comment · Reviewer_z7aM · 2025-11-25
> > **Scores have been raised**
> >
> > Thank you for the detailed experiments and analysis, which have addressed most of my concerns. After careful consideration, I will raise my score properly.

---

> > > ### Author Response · Authors · 2025-11-27
> > >
> > > Thank you again for your insightful and constructive review. We are delighted that our response addressed your concerns and appreciate you raising your score.

---

### Author Response · Authors · 2025-11-23
**Global Response**

We sincerely thank the reviewers for their constructive feedback and for recognizing the value of our work. We are encouraged that the reviewers found our paper to have **"innovative insights"** (Reviewer v2vp) and **"solid experimental quality"** (Reviewer v2vp), while providing a **"clear diagnosis"** of modality conflicts (Reviewer xf1h). We also appreciate the recognition of our **"clear presentation"** (Reviewer z7aM) and the contribution of our **open-source code** (Reviewer z7aM).

We have carefully addressed all concerns and updated the paper accordingly. Below is a summary of our responses to common questions.

> **1. Clarification and Validation of Gradient Conflict Metric (Respond to z7aM, v2vp, 7qva, xf1h)**


All reviewers raised questions regarding the definition and validity of the gradient conflict metric ($c_g$).
* **Definition & Rationale:** We have rewritten **Section 3.1** and added a illustration to clarify the calculation.
* **Empirical Validation:** To prove that our metric reflects real-world optimization dynamics, we conducted a quantitative analysis on mainstream LLMs (Qwen, Llama). We found that the gradient conflict between **Code vs. Math** is strictly lower than **Code vs. Wiki**. This correlates perfectly with established findings that Code and Math training mutually enhance each other (positive transfer). Furthermore, our experiments show that Qwen2.5-Coder (fine-tuned on code) achieves substantial gains in Math, confirming that **low gradient conflict correlates with positive transfer**, validating our metric's effectiveness.

> **2. Expanded Evaluation: New Benchmarks and Scaling (Respond to z7aM, v2vp)**


Reviewers requested more comprehensive comparisons on image generation and editing tasks. We have added the following evaluations:

* **Image Generation:** We added **MSCOCO** and **T2I-CompBench**. Uni-X (3B/4.5B) achieves a T2I-Avg score of **66.6** and MSCOCO CLIP-T score of **31.8**, outperforming larger autoregressive baselines like Liquid (7B) and EMU3 (8B).
* **Image Editing:** We evaluated Uni-X on **ImgEdit**. Despite using significantly less training data (90k samples) and fewer parameters, Uni-X achieves an overall score of **3.44**, surpassing **Bagel (7B/14B, score 3.20)** and approaching specialized editing models.
* **Scaling:** We trained a scaled version of Uni-X (Uni-X$^{\heartsuit}$) on 240B tokens. It achieves SOTA performance among AR UMMs without semantic encoders, demonstrating strong scalability.

> **3. Inference Efficiency Analysis (Respond to xf1h)**

We addressed the concern regarding the inference efficiency of the split architecture.
* **Theoretical Speedup:** In the separated layers, attention complexity reduces from $O((a+b)^2)$ to $O(a^2 + b^2)$ due to strict modality isolation.
* **Empirical Throughput:** We measured throughput on an H800 PCIE GPU (350W). Uni-X achieves **910.2 tokens/s**, which is comparable to the fully shared baseline (975.2 tokens/s) and significantly faster than **EMU3 (199.0 tokens/s)** and **Liquid (182.0 tokens/s)**. Notably, Uni-X is **~18x faster** than EMU3 in image generation tasks due to our efficient tokenization strategy.

> **Summary of Paper Updates**

1.  Clarified the mathematical definition and rationale of the Gradient Conflict metric, and added an illustration to intuitively introduce the concept of gradient.
2.  Rewrote the section on gradient conflict in the introduction, introducing the impact of gradient conflict on model optimization.
3.  Added image editing performance to the main text, better demonstrating the superiority of Uni-X.
4.  Added new appendix experiments:
    *   Added analysis on Code/Math/Wiki domains to validate the metric (Table R1, R2 in individual responses).
    *   Added results for MSCOCO and T2I-CompBench.
    *   Added inference throughput comparisons.
    *   Added ablation studies on the ratio of Text/Vision layers.

---

### Author Response · Authors · 2025-11-29
**Discussion Summary for AC (1/2)**

Dear AC,

Thank you for coordinating the review process. We would like to briefly summarize the rebuttal phase for you here.

Reviewer xf1h highly recognized our work, gave an initial score of 8, and replied on November 24th that their main concerns have been addressed. Although the other three reviewers gave negative ratings, **all three reviewers demonstrated a willingness to raise their scores**.

* Reviewer z7aM stated our rebuttal **resolved his concerns and raised the score to 6** (on **November 25th**, prior to the widespread information leak bug on November 27th; this is for your reference).
* Reviewers v2vp and 7qva both **acknowledged our contribution and novelty**, mainly having some misunderstandings regarding the concept of gradient conflict. We resolved this through a **detailed illustration added to paper** and a more concrete definition.
  * Specifically, Reviewer v2vp stated: *"The paper presents innovative insights, with solid experimental quality and clear expression, and holds significant importance."*
  * Reviewer 7qva stated: *"In summary, my primary concerns lie in Weaknesses 1 (definition of gradient conflict) and Weaknesses 2 (performance drop on text-only benchmarks). If the responses are reasonable and acceptable, I will raise the score."*

Next, we summarize how we addressed their concerns **point-by-point through experiments**.

> **1. Clarification and Empirical Validation of Gradient Conflict Metric (Respond to z7aM[Q1], v2vp[W1, Q1], 7qva[W1, Q1], xf1h[W1, Q1])**

All reviewers raised questions regarding the definition and validity of the gradient conflict metric ($c_g$).
* **Definition:** We have rewritten Section 3.1 and **added an illustration (Figure 2)** to clarify.
* **Empirical Validation:** To prove that our metric reflects real-world optimization dynamics, we conducted a **quantitative analysis on mainstream LLMs (Qwen, Llama)**. We found that the gradient conflict between **Code vs. Math** is strictly lower than **Code vs. Wiki**. This correlates perfectly with established findings that Code and Math training mutually enhance each other (positive transfer) [1, 2]. Furthermore, our experiments show that Qwen2.5-Coder (fine-tuned on code) achieves substantial gains in Math, confirming that **low gradient conflict correlates with positive transfer**, validating our metric's effectiveness.

> **2. Expanded Evaluation: New Benchmarks and Scaling (Respond to z7aM[W1, W2, W3], v2vp[W3])**

* **Image Generation:** We added **T2I-CompBench** and **MSCOCO**. Uni-X (3B/4.5B) achieves a T2I-Avg score of **66.6** and MSCOCO CLIP-T score of **31.8**, outperforming larger autoregressive baselines like Liquid (7B) and EMU3 (8B).
* **Image Editing:** We evaluated Uni-X on **ImgEdit**. Despite using significantly less training data (90k samples) and fewer parameters, Uni-X achieves an overall score of **3.44**, surpassing **Bagel (7B/14B, score 3.20)** and approaching specialized editing models.
* **Scaling:** We trained a scaled version of Uni-X (Uni-X$^{\heartsuit}$) on 240B tokens. It achieves **SOTA performance among AR UMMs without semantic encoders**, demonstrating strong scalability.

> **3. Similarities and Differences between Uni-X and Classical Dual-Tower Structure (Respond to v2vp[W2, Q2])**

We compared them from three perspectives. First, regarding **Middle-Sharing for Semantic Fusion**, we supported this via our ablation study (Table 5 of main paper) and answered Reviewer v2vp's Q2, clarifying that the classical dual-tower structure is not more suitable for unified understanding and generation. We also argued from the perspectives of **Different Objectives** and **Single AR Framework**.

> **4. Lack of Comparison with Recent Model Blip3o-Next (Respond to v2vp[W3])**

**First, we emphasize that Blip3o-Next was uploaded to arXiv on October 17, while the ICLR submission deadline was September 24.**
Secondly, we also compared them through experiments. Uni-X's performance was slightly inferior. However, we believe this result is acceptable:
* Blip3o-Next used GenEval as a reward for reinforcement learning, which might lead to an inflated score on GenEval.
* Blip3o-Next used 10 million samples for training on image editing tasks, while Uni-X only used 90k, yet still achieved comparable performance.
* Blip3o-Next uses a pre-trained diffusion backbone SANA-1.5, which is equivalent to performing additional training on image generation tasks.

---

> ### Author Response · Authors · 2025-11-29
> **Discussion Summary for AC (2/2)**
>
> > **5. Performance Drop of Uni-X on Text-Only Benchmarks (e.g., MMLU, ARC) (Respond to 7qva[W2])**
>
> We directly proved through experiments that: **this is because the quality of our pre-training text data is inferior to Qwen's proprietary annealing data**.
>
> We verified this through a controlled experiment: we performed continual training on the original Qwen2.5-3B model using **only the text component** of the Uni-X pre-training dataset (completely removing image modality data), **which reproduced the performance decline observed in Uni-X.**
>
> > **6. Ablation Study on Hyperparameter "Ratio between t-layers and v-layers within the separated layers" (Respond to 7qva[Q2])**
>
>
> We conducted experiments to further refine the paper. Results show that reducing the number of vision layers also decreases pure text performance. This suggests that the shared layers in the middle have to process more low-level vision information when vision layers are reduced, thereby impacting pure text capability. This result further validates the effectiveness of our proposed architecture.
>
> ---
>
> The above summarizes our response. We believe we have resolved the reviewers' concerns through experiments and clarifications. We hope the AC will consider these positive developments during the discussion period when making the final decision.
>
> Best regards,
>
> Authors
>
> [1] At Which Training Stage Does Code Data Help LLMs Reasoning?
>
> [2] DeepSeekMath: Pushing the Limits of Mathematical Reasoning in Open Language Models

---

### Meta-Review · Area_Chair_yNvA · 2025-12-15

**Summary:**

The paper introduces Uni-X, a unified multimodal model architecture designed to mitigate gradient conflict in shared autoregressive Transformers. Uni-X adopts an X-shaped design which separates modality-specific layers at both ends while sharing intermediate layers to reduce conflicts between vision and text modalities. Experiments show Uni-X scales efficiently, achieves competitive performance in generation and understanding tasks, and offers a parameter-efficient alternative to larger baselines. Uni-X offers a promising and elegant approach to unified multimodal modeling, backed by solid experiments and scalability benefits. However, the paper would benefit from clearer explanations of gradient conflict, broader evaluations, and deeper analysis of architectural trade-offs. The authors did a good job on the rebuttal and addressed most of concerns.

**Reviewer Concerns:**

I think most of concerns have been addressed by the rebuttal.

**Reviewer Scores:**

I think the reviewers would have changed their scores.

---

### Decision · Program_Chairs · 2026-01-26

Accept (Poster)